# PATH AUXILIARY PROPOSAL FOR MCMC IN DISCRETE SPACE

**Haoran Sun**
Georgia Institute of Technology
hsun349@gatech.edu

**Hanjun Dai**
Google Brain
hadai@google.com

**Wei Xia**[*]
Amazon
weixxia@amazon.com

**Arun Ramamurthy**
Siemens
arun.ramamurthy@siemens.com

## ABSTRACT

Energy-based Models (EBMs) offer a powerful approach for modeling discrete structure, but both inference and learning of EBM are hard as it involves sampling from discrete distributions. Recent work shows Markov Chain Monte Carlo (MCMC) with the informed proposal is a powerful tool for such sampling. However, an informed proposal only allows local updates as it requires evaluating all energy changes in the neighborhood. In this work, we present a path auxiliary algorithm that uses a composition of local moves to efficiently explore large neighborhoods. We also give a fast version of our algorithm that only queries the evaluation of energy function twice for each proposal via linearization of the energy function. Empirically, we show that our path auxiliary algorithms considerably outperform other generic samplers on various discrete models for sampling, inference, and learning. Our method can also be used to train deep EBMs for high dimensional discrete data.

## 1 INTRODUCTION

Many real-world problems involve discrete structured data modeling, such as syntax trees for natural language processing(Tai et al., 2015), graphical model for molecules(Gilmer et al., 2017), etc. A powerful approach for modelling the distribution over structured data is Energy Based Models(LeCun et al., 2006) (EBMs). EBMs define the distribution with an unnormalized energy function, which allows great flexibility to fit the target distribution. However, this flexibility also results in the difficulties in inference and learning, as they require sampling from the EBM (Andrieu et al., 2003; Hinton, 2002), where the partition function is intractable in many cases.

Markov Chain Monte Carlo (MCMC) algorithms are one of the most widely used methodologies to sample from intractable distributions (Robert & Casella, 2013). The efficiency for MCMC depends drastically on the proposal distribution. For example, in continuous space, Metropolis-adjusted Langevin algorithm (MALA) exploits the gradient of the target in single step walk and biases the proposal distribution toward high probability region (Rossky et al., 1978; Roberts & Rosenthal, 1998; Welling & Teh, 2011); Hamiltonian Monte Carlo (HMC) employs multi-step walk and explores the distribution more efficiently(Neal, 2004; Girolami & Calderhead, 2011; Hoffman et al., 2014). These methods substantially improve the performance of the MCMC algorithm in theory and in practice. However, their proposal distributions are derived as discretization of continuous diffusion process and it is still not clear how to appropriately extend such methods into discrete space(Zanella, 2020).

Recently, Zanella (2020) proposed a general framework called pointwise informed proposal (PIP) that shows promising results on directly sampling from discrete distributions. PIP utilizes the energy change in the neighborhood of current state to propose a new state. Following this work, Grathwohl et al. (2021) propose a more efficient sampler that uses Taylor series to estimate the energy change

---

[*]Work done during the time at Siemens

in the neighborhood. However, both methods only focus on proposing from a small neighborhood, for example, a 1-Hamming ball. This is due to the computational expense of evaluating the energy change or approximation error of Taylor series for a large neighborhood. As a result, the samples in Markov chain will have strong correlation or even be trapped at local optimum, which deteriorates the sampling efficiency.

In order to efficiently explore a larger neighborhood, we propose a path auxiliary sampler, which is an auxiliary sampler (Liang et al., 2011) that uses auxiliary path to propose new states. In construction of the path, we employ a local proposal distribution to make a sequence of small movements to reach a new state in long distance. A typical challenge for such multi-step proposal is that the accept ratio could decrease very fast when the number of steps increases. In this work, we show that the accept ratio of our path auxiliary proposal is *independent* of the path and only determined by the property of the current state and the proposed state. As a result, our algorithm is able to maintain a high accept ratio with multiple local steps. We can provably show that the locally balancing functions are asymptotically optimal for our path auxiliary sampler. We also introduce a scalable version of the algorithm that uses linearization and can be applied to smooth target distributions.

We empirically evaluate our methods in inference, sampling, and learning on various discrete EBMs. We demonstrate that our methods significantly improve the sampling efficiency on parity model, weighted permutation model, Ising model, Restricted Boltzmann Machine, and factorial Hidden Markov Model. Our method can also learn competitive deep EBMs on discrete image data. The code can be found at `https://github.com/ha0ransun/Path-Auxiliary-Sampler.git`.

## 2 BACKGROUND

**Energy Based Model**: Let $\mathcal{X}$ be a finite state space. An EBM defines an energy function $f : \mathcal{X} \to \mathbb{R}$ with target distribution $\pi(x) = e^{-f(x)}/Z$, where $Z = \sum_{z \in \mathcal{X}} e^{-f(z)}$ (LeCun et al., 2006; Wainwright & Jordan, 2008; Du & Mordatch, 2019). The unnormalized energy function provides great flexibility to characterize complex distribution. However, this can also make the partition function $Z$ intractable to calculate and exacerbate the difficulties of learning and inference.

**Metropolis-Hastings**: MCMC is a commonly used methodology to sample from an intractable distribution. Metropolis-Hastings (MH) is one of the most commonly used framework for MCMC(Metropolis et al., 1953; Hastings, 1970). Given the current state $x$, a proposal distribution $Q(x, \cdot)$ gives a new state $y$, then MH algorithm accept $y$ with probability of $\min\{1, \frac{\pi(y)Q(y,x)}{\pi(x)Q(x,y)}\}$ to guarantee the Markov chain is $\pi$-reversible. The efficiency of MH algorithm highly depends on the selection of proposal distribution $Q$.

**Peskun Ordering**: Peskun ordering provides a method to compare the efficiency for two MCMC algorithms (Peskun, 1973; Tierney, 1998). Let $P_1, P_2$ be $\pi$-reversible Markov transition kernels on $\mathcal{X}$ such that $P_1(x, y) > cP_2(x, y)$ for all $x \neq y$ for a fixed $c > 0$, then we say $P_1$ is $c$-times more efficient than $P_2$ as the following holds: 1) $\text{var}_\pi(h, P_1) \leq \frac{1}{c}\text{var}_\pi(h, P_2) + \frac{1-c}{c}\text{var}_\pi(h), \quad \forall h : \mathcal{X} \to \mathbb{R}$; 2) $\text{Gap}(P_1) \geq c\,\text{Gap}(P_2)$. A smaller *asymptotic variance* $\text{var}_\pi(h, P)$ means a better estimation for the expectation of $h$ and a larger *spectral gap* $\text{Gap}(P)$ means a faster convergence of the Markov chain.

**Pointwise Informed Proposals and Locally Balancing Function**: A pointwise informed proposal (PIP) is a MH algorithm in discrete space (Zanella, 2020). PIP uses proposal distribution $Q_g(x, y) = g(\frac{\pi(y)}{\pi(x)})I(x, y)/Z_g(x)$, where the $I(x, y) = 1_{\{y \in \mathcal{N}(x)\}}$ is the membership indicator w.r.t a symmetric neighborhood $\mathcal{N}(\cdot)$, and the normalizer $Z_g(x) = \sum_{z \in \mathcal{X}} g(\frac{\pi(z)}{\pi(x)})I(x, z)$. The weight function $g : \mathbb{R}_+ \to \mathbb{R}_+$ determines the efficiency of PIP. Zanella (2020) show that the family of locally balancing functions $\mathcal{G} = \{g : \mathbb{R}_+ \to \mathbb{R}_+, g(t) = tg(\frac{1}{t}), \forall t > 0\}$ is asymptotically optimal for PIP w.r.t. Peskun ordering. Empirically, Zanella (2020) shows $g(t) = \sqrt{t}$ and $g(t) = \frac{t}{t+1}$ have the best performance.

**Gibbs with Gradient** GWG (Grathwohl et al., 2021) is a scalable version of PIP, where the target distribution $\pi(\cdot)$ is approximated via Taylor's series. Despite being powerful, PIP and GWG requires the calculate the weight and sample from the neighborhood. Hence, only a small neighborhood, usually a 1-Hamming ball, is used in existing methods.

## 3 PATH AUXILIARY PROPOSAL

Problems occur in point-wise informed proposal (PIP) when merely considering a small neighborhood, especially for distributions with many local optima. For example, consider a parity distribution where the state space $\mathcal{X} = \{0,1\}^p$, and the energy function $f(x) = U$ if the number of 1s in $x$ is odd, otherwise $f(x) = 0$. When a 1-Hamming ball neighborhood is used, by symmetry, a PIP will have a uniform probability to propose a new state from neighborhood. Then, the expected time to leave a low energy state is $O(e^U)$, which could be very inefficient when $U$ is large. When a $r$-Hamming ball larger neighborhood is used, a PIP can efficiently escape a low energy state. However, the neighborhood will contain $O(n^r)$ states, which could be computationally prohibitive when $n$ or $r$ is large.

To address such problems with PIP, we propose a path auxiliary sampler. Instead of directly sampling from a large neighborhood, path auxiliary sampler sequentially samples new state from a local proposal distribution $Q_0$. The computation cost at each step is still manageable as $Q_0$ still samples locally. Then the composition of small movements forms a path that can lead to a new state that is distant from the current state. The complexity of the sampling grows linearly instead of exponentially w.r.t. $r$. In this section, we first present the framework of our algorithm. Then, we discuss the choice of the weight function for path auxiliary sampler. And finally, we introduce a fast version of path auxiliary sampler.

### 3.1 PROPOSAL VIA AUXILIARY PATH

We first define an auxiliary path. Given the neighborhood function $\mathcal{N}$, the set of auxiliary paths on $\mathcal{X}$ is defined as

$$\Sigma(\mathcal{X}, \mathcal{N}) := \{(\sigma, L) : \sigma_i \in \mathcal{X}, i = 0, ..., L; \sigma_l \in \mathcal{N}(\sigma_{l-1})), l = 1, ..., L\} \tag{1}$$

To obtain the auxiliary path, we employ a PIP $Q_0(\cdot, \cdot)$ to make local movements. We also sample path length from a prior $\alpha(\cdot)$ to assure our chain is aperiodic. The path auxiliary sampler is defined as follows

1. Sample a path length $L$ from a prior $\alpha(L)$.

2. Denote the current state $x_t = \sigma_0$, sample $\sigma_l \sim Q_0(\sigma_{l-1}, \cdot)$ for $l = 1, ..., L$.

3. Accept $x_{t+1} = \sigma_L$ in probability $A(x, \sigma, L) = \min\left\{1, \frac{\pi(\sigma_L) \prod_{l=1}^{L} Q_0(\sigma_l, \sigma_{l-1})}{\pi(\sigma_0) \prod_{l=1}^{L} Q_0(\sigma_{l-1}, \sigma_l)}\right\}$, else $x_{t+1} = x_t$.

**Theorem 1.** *The path auxiliary proposal transition rule described above satisfies the detailed balance and induces a reversible Markov chain with $\pi$ as its invariant distribution.*

### 3.2 BENEFIT FROM AUXILIARY PATH

We first analyze the performance of path auxiliary sampler on parity distribution mentioned above. Assume we use a uniform prior $\alpha(1) = \alpha(2) = \frac{1}{2}$. If we sample $L = 2$, then we will transit back to a new state with same energy as the current state. As a result, we can always escape from a state in $O(1)$ steps in expectation.

Having a path auxiliary sampler to sample from $L$-Hamming ball at each step can be more efficient than performing $L$ steps of MH sampling in a 1-Hamming ball. To mathematically justify it, we compare their accept ratio. For a path $(\sigma, L)$:

$$\min\left\{1, \frac{\pi(\sigma_L) \prod_{l=1}^{L} Q_0(\sigma_l, \sigma_{l-1})}{\pi(\sigma_0) \prod_{l=1}^{L} Q_0(\sigma_{l-1}, \sigma_l)}\right\} \geq \prod_{l=1}^{L} \min\left\{1, \frac{\pi(\sigma_l) Q_0(\sigma_l, \sigma_{l-1})}{\pi(\sigma_{l-1}) Q_0(\sigma_{l-1}, \sigma_l)}\right\} \tag{2}$$

We can notice that the accept ratio for path auxiliary sampler is the product of the probability for single-step method before the minimum operator. As a result, an auxiliary path can have a high accept ratio as long as the product is large. On the contrary, when performing $L$ steps MH sampling, the transition can probably be blocked at some steps in the path with low accept ratios.

### 3.3 BALANCED PROPOSAL

Although the product in equation 2 allows the Markov chain to escape from a local optimum when some steps in the path have high accept ratio, it can also exponentially decrease the accept ratio w.r.t. the path length $L$ when every step has a low accept ratio. Hence, it is important to select a good local proposal $Q_0$. In this section, we show that the locally balancing function Zanella (2020) is asymptotically optimal for path auxiliary sampler. We first define a sub-class of weight function named as ideal function:

**Definition 1.** $\mathcal{F}$ *is the set of ideal function. A function* $f \in \mathcal{F}$ *if and only if following conditions holds: 1)* $f : \mathbb{R}_+ \to \mathbb{R}_+$*; 2)* $f(1) = 1$*; 3)* $f(t)$ *is monotonic increasing;4)* $f(t)f(\frac{1}{t})t \le 1, \forall t \le 1$

The next theorem shows that locally balancing function $\mathcal{G}$ is asymptotically better than ideal function $\mathcal{F}$ in Peskun Ordering.

**Theorem 2.** *Consider the state space in Cartesian products* $\mathcal{X} = \times_{i=1}^n \mathcal{X}_i$*, where each* $\mathcal{X}_i$ *is a finite space with $M$ elements, and the neighborhood is defined as 1-Hamming ball. Let $d_n$ be the maximum degree in conditional independence graph. If 1)* $\lim_{n \to \infty} \frac{d_n}{n} = 0$*; 2) the target distribution satisfies* $\frac{\pi(y)}{\pi(x)} \le C < \infty, \forall y \in \mathcal{N}(x)$*; 3) the path length prior $\alpha$ is bounded by $U$. Then for any* $f \in \mathcal{F}$*, we have a function* $g \in \mathcal{G} = \{g(\cdot) : g(t) = tg(\frac{1}{t})\}$ *that is asymptotically more efficient than $f$, which implies $\mathcal{G}$ is asymptotically optimal in $\mathcal{F}$.*

The idea to prove this theorem is to use $\bar{f}(t) := \sqrt{f(t)f(\frac{1}{t})t}$. By definition, we have:

$$\bar{f}(t) = \sqrt{f(t)f(\frac{1}{t})t} = t\sqrt{f(t)f(\frac{1}{t})\frac{1}{t}} = t\bar{f}(\frac{1}{t}) \tag{3}$$

which means $\bar{f} \in \mathcal{G}$. Then, the theorem is proved by showing the $\pi$-reversible Markov transition kernel $P_{\bar{f}}(x, y) \ge P_f(x, y)$ asymptotically holds. The proof is given in Sec A.5

Although Definition 1 has some constraints, it already contains almost all natural choices of weight functions, such as $f(t) = \frac{2t}{1+t}$, $f(t) = \min\{1, t\}$, $f(t) = \max\{1, t\}$, and $f(t) = t^\alpha, \alpha \ge 0$. Hence, the asymptotic optimality for locally balanced function $\mathcal{G}$ in ideal function $\mathcal{F}$ strongly suggest that locally balanced function is a good choice for path auxiliary sampler. More discussion for ideal function can be found in Sec A.3

**Property 1.** *When using a locally balancing function $g \in \mathcal{G}$ as the weight function, the accept ratio for path auxiliary sampler can be written as:*

$$A(x, y, \sigma, L) = \min\left\{1, \frac{Z_g(x)}{Z_g(y)}\right\} \tag{4}$$

The path independent form in Property 1 shows that if the energy function is smooth enough, the accept ratio in equation 4 will be high when $y$ is close to $x$. By selecting $g(t) = \sqrt{t} \in \mathcal{G}$, we name our auxiliary sampler (Liang et al., 2011) as path auxiliary sampler (PAS) and give the algorithm in Algorithm 1.

### 3.4 SCALABLE PATH AUXILIARY ALGORITHM

Although path auxiliary proposal reduces the complexity from exponential to linear, it still requires $O(nL)$ evaluations of the probability $\pi(x)$ in total. In cases where the computational bottleneck is evaluating $\pi(x)$, this cost will be expensive. Fortunately, most distributions of interest have differentiable energy functions, such as deep EBMs , and we can use the linearization as approximation (Grathwohl et al., 2021). Given the current state $\sigma_0$, the linearization is

$$\tilde{f}_{\sigma_0}(z) = f(\sigma_0) + \langle \nabla f(\sigma_0), z - \sigma_0 \rangle \tag{5}$$

we can estimate $\tilde{\pi}_{\sigma_0}(y)/\tilde{\pi}_{\sigma_0}(x) = e^{-\langle \nabla f(\sigma_0), y-x \rangle}$, and define our proposal distribution $\tilde{Q}_{g,\sigma_0}(x, y) = g(\frac{\tilde{\pi}_0(y)}{\tilde{\pi}_0(x)})/\tilde{Z}_{g,\sigma_0}(x)$, with the normalizer $Z_{g,\sigma_0}(x) = \sum_{z \in \mathcal{N}(x)} g(\frac{\tilde{\pi}_0(z)}{\tilde{\pi}_0(x)})$. After sampling $\sigma$, we use the linearization $\tilde{f}_{\sigma_L}$ at $\sigma_L$ to compute the accept ratio w.r.t. the auxiliary path

$$A(x, \sigma, L) = \min\left\{1, \frac{\pi(y) \prod_{l=1}^L \tilde{Q}_{g,\sigma_L}(\sigma_l, \sigma_{l-1})}{\pi(x) \prod_{l=1}^0 \tilde{Q}_{g,\sigma_0}(\sigma_{l-1}, \sigma_l)}\right\} \tag{6}$$

---

**Algorithm 1:** Path Auxiliary Sampler (PAS) and the fast version (PAFS)

---

**Input:** Target Distribution $\pi$, Initial state $x_0$, Path Length Prior $\alpha$, Weight Function $g(t) = \sqrt{t}$
**Output:** Sample sequence $(x_1, x_2, ...)$

1 **repeat**
2      Sample path length $L \sim \alpha(L)$
3      Denote $\sigma_0 = x_t$, sample $\sigma_l \sim Q_g(\sigma_{l-1}, \cdot)$ or $\sigma_l \sim \tilde{Q}_{g,\sigma_0}(\sigma_{l-1}, \cdot)$ for $l = 1, ..., L$.
4      Accept $x_{t+1} = \sigma_L$ in probability $A(x, \sigma, L) = \min\{1, \frac{Z_g(x_t)}{Z_g(\sigma_L)}\}$
5          or $A(x, \sigma, L) = \min\left\{1, \frac{\pi(y)\prod_{l=1}^{L}\tilde{Q}_{g,\sigma_L}(\sigma_l, \sigma_{l-1})}{\pi(x)\prod_{l=1}^{0}\tilde{Q}_{g,\sigma_0}(\sigma_{l-1}, \sigma_l)}\right\}$, else $x_{t+1} = x_t$.
6 **until** *finish sampling*;

---

By using approximation, we only need to evaluate $\pi(y), \pi(x), \nabla f(y), \nabla f(x)$ once in every proposal, which can significantly reduce the computational cost. We name this algorithm as path auxiliary fast sampler (PAFS) (see Algorithm 1).

The PAS framework also allows different approximations. For example, GWG (Grathwohl et al., 2021) can be seen as a special case where all indices are sampled from $\tilde{Q}_{g,\sigma_0}(\sigma_0, \cdot)$ rather than $\tilde{Q}_{g,\sigma_0}(\sigma_l, \cdot)$. Besides, one way to further encourage long range movements is sampling sites without replacement, that's to say, the auxiliary path $\sigma$ does not modify a site more than once. However, such a sampler requires a decent choice of the path length $L$. When $L$ is too large, the acceptance rate drops exponentially fast. Hence, in this work, we focus on PAFS. Similar to GWG (Grathwohl et al., 2021), the decrease of the proposal quality from PAFS can be bounded.

**Theorem 3.** *Assume the energy function $f(x)$ is differentiable, $\nabla f(x)$ is $K$-Lipschitz. Consider we use weight function $g(t) = \sqrt{t}$, path length prior $\alpha$ bounded by $U$, and 1-Hamming ball as neighborhood. Denote $P$ and $\tilde{P}$ as the transition kernel by path auxiliary sampler and path auxiliary faster sampler, respectively, we have*

$$\tilde{P}(x, y) \geq e^{-K\frac{U(U+1)}{2}}P(x, y) \tag{7}$$

Using the Peskun ordering, we know PAFS is at least $e^{-K\frac{U(U+1)}{2}}$ times as efficient as PAS. When $K$ is small, theorem 3 justify PAFS is a good approximation for PAS. When $K$ is large, a tighter bound needs more assumptions for the target distribution, e.g. conditional independence. See the proof and discussion in Sec A.7.

## 4 RELATED WORKS

Informed proposal for Metropolis-Hastings (MH) algorithm has been extensively studied in the continuous space(Robert & Casella, 2013). The most famous algorithms are Metropolis-adjusted Langevein algorithm (MALA) (Roberts & Rosenthal, 1998) and Hamiltonian Monte Carlo (HMC). MALA, HMC, and their variants (Girolami & Calderhead, 2011; Hoffman et al., 2014; Welling & Teh, 2011; Titsias & Papaspiliopoulos, 2018) exploit the gradient of the target distribution to bias the proposal distribution towards high probability regions. Although gradient-based methods having brought substantial improvements in continuous space, it is still unclear how to extend them to discrete space.

A number of methods try to map the discrete space to a continuous space using relaxation, apply informed methods in continuous space, and then map the new state back into discrete space(Zhang et al., 2012; Pakman & Paninski, 2013; Nishimura et al., 2017; Han & Liu, 2018; Jaini et al., 2021) via Gaussian Integral Trick, uniform dequantization, or VAE flow. Such methods work in some scenarios, but a key challenge is the embedding of discrete space into continuous space can destroy the inherent discrete structure, resulting in highly multi-modal and irregular target distribution in continuous spaces.

Another group of methods directly work on discrete space. Dai et al. (2020) introduces the path as latent variable in the variational distribution for initializing PCD, but still relies on slow Gibbs sampling for improvement; Titsias & Yau (2017) augment the discrete space with auxiliary variable and

perform Gibbs sampling in the augmented space based on informed proposal. Zanella (2020) shows that a family of locally balancing function is asymptotically optimal for informed proposal. Following Zanella (2020), Power & Goldman (2019) extends the framework to Markov jumping process, and Sansone (2021) parameterize locally balanced function to tune it via mutual information to select good weight function from the locally balanced class. When the target distribution is smooth enough, Grathwohl et al. (2021) employs a Taylor's series to approximate the target distribution and further improve the sampling efficiency. Though these informed proposal algorithms successfully show orders of magnitude improvements when using 1-Hamming ball as neighborhood, they are not able to explore a large neighborhood. An extension of GWG (Grathwohl et al., 2021) can partially address this problem via sampling multiple dimensions to modify in one step. However, such procedure can easily lead to backtracks thereby reducing its efficiency.

## 5 EXPERIMENTS

### 5.1 SAMPLERS UNDER CONSIDERATION

In this section, we empirically evaluate the sampling efficiency of path auxiliary sampler (PAS) and path auxiliary fast sampler (PAFS). We choose the path length prior $\alpha$ as a uniform distribution on $\{1, ..., 2X - 1\}$ as suggested in Hoffman et al. (2021) and denote the corresponding samplers as PAS-$X$ and PAFS-$X$.

We compare our methods with five types of baselines: random walk sampler (RW), Gibbs sampler (Gibbs), Hamming ball sampler (HB), locally balanced sampler (LB), Gibbs with gradient sampler (GWG). RW is an informed proposal with $g(t) = 1$ that uniformly propose new state from neighborhood. Gibbs partitions the dimension of a state $x$ into two groups $x_u$ and $x_{-u}$, then updates the state from conditional distribution $p(x_u|x_{-u})$. We denote it as Gibbs-X, where X refers to the dimension of $x_u$. HB is a two-stage Gibbs sampler on the extended state space $(x, x')$ (Titsias & Yau, 2017). We use HB-10-1, where 10 is the block size, and 1 is the hamming ball size. LB (Zanella, 2020) is implemented as a single-step PIP with $g(t) = \sqrt{t}$. GWG (Grathwohl et al., 2021) is a scalable version of LB, which draw the index to flip via first order Taylor's series of the target distribution. Although the original paper focuses on flipping one site per step, it is possible for GWG to draw multiple indices to flip in every step. We denote the algorithm as GWG-X, where X indicates that the number of indices to modify per step is uniformly sampled from $\{1, ..., 2X - 1\}$. The difference between PAFS and GWG is that GWG is more likely to sample the same index repeatedly thereby reducing the efficiency especially after mixing, e.g in figure 6.

### 5.2 INFERENCE ON ENERGY BASED MODEL

**Parity Model:** We first demonstrate the benefit of path auxiliary sampler versus single-step sampler in a inference task that estimates the mean $\mu$ of a parity distribution. A parity distribution has state space $\mathcal{X} = \{0, 1\}^p$ and energy function

$$f(x) = ((\sum_{i=1}^{p} x_i) \bmod 2)U \tag{8}$$

i.e. a state has energy $U$ if it has an odd number of 1s, otherwise 0. The neighborhood is defined as 1-Hamming ball. We run the simulation with $p = 100$ and $U = \{1, 3, 5\}$, and report the estimation

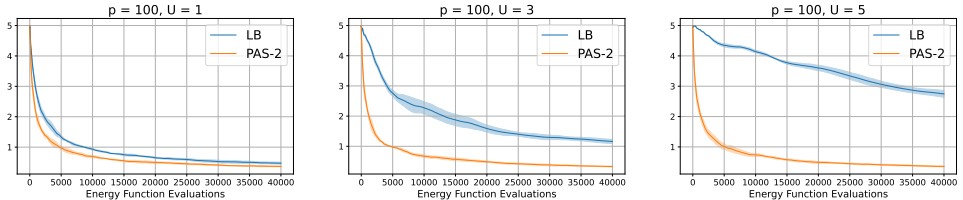

Figure 1: Estimation Error of Distribution Mean on Parity Model

error $E_n := \|\hat{\mu}_n - \mu\|_2$, where $\hat{\mu}_n = \sum_{i=1}^{n} X_i$ is the sample mean of the Markov chain at step $n$. For each setting and method, we run 5 chains for 20,000 steps, and we plot the mean and standard deviation of the estimation error in figure 1. We can observe that the efficiency of single step sampler

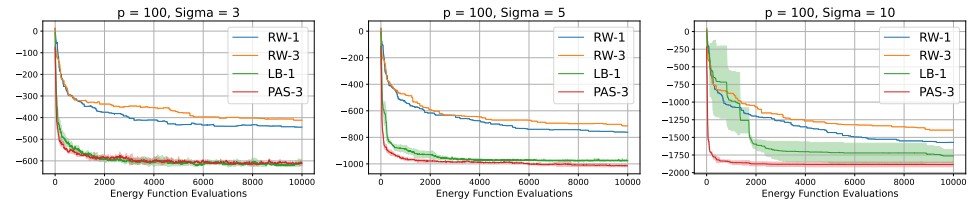

Figure 2: Energy Trace on Weighted Permutation Model

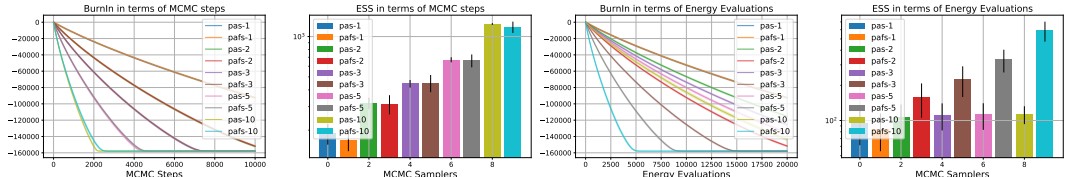

Figure 3: PAS and PAFS in different lengths on $200 \times 200$ Ising Model

decreases exponentially when $U$ increases. On the contrary, the path auxiliary sampler can always escape from the local optima in parity distribution and estimate the distribution mean efficiently.

**Weighted Permutation Model:** We consider an optimization task on weighted permutation model. The state space $\mathcal{X} = S_p$ is a symmetric group, i.e. for any $\rho \in \mathcal{X}$, $\rho$ is a permutation of $1, 2, ..., p$. The energy function is defined as:

$$f(\rho) = \sum_{i=1}^{p} w_{i,\rho(i)} \tag{9}$$

Following Zanella (2020), the weight $\{w_{ij}\}_{i,j=1}^{p}$ are i.i.d. sampled from Gaussian$(0, \sigma^2)$. The neighborhood is determined by one exchange of the permutation. We use $p = 100$, $\sigma = 3, 5, 10$, and run 5 simulations for each configurations. From figure 2, we can see that LB sampler is trapped at local optimum, especially when $\sigma$ is large and the distribution is sharp. On the contrary, PAS can always efficiently find good solutions via auxiliary path.

### 5.3 SAMPLING ON ENERGY BASED MODEL

**Lattice Ising Model:** Consider the state space $\mathcal{X} = \{-1, 1\}^{V_p}$, where $(V_p, E_p)$ is the $p \times p$ square lattice graph. For each $x \in \mathcal{X}$, the energy function is defined as:

$$f(x) = -\sum_{i \in V_p} \alpha_i x_i - \lambda \sum_{(i,j) \in E_p} x_i x_j \tag{10}$$

$\alpha_i \in \mathbb{R}$ are bias terms representing the property of $x_i$ and $\lambda$ is a global interaction term. We first compare PAS and PAFS in different path lengths. For each length, we run 100 chains with 100,000 steps. We report the burn-in stage as well as effective sample size (ESS) [*] in terms of both MCMC steps and energy function evaluation times in figure 3. We can see that, 1) PAS has slight improvement in terms of energy function evaluations when we increase the path length; 2) when compared to PAS, PAFS has very similar proposal quality and significantly better efficiency. We also compare our sampler with other competitors. For each sampler, we run 5 Markov chains with 1,000,000 steps and report the ESS to compare the proposal quality in figure 4. We can see our sampler leads in both quality and efficiency. More results are given in Sec B.1.

**factorial Hidden Markov Model:** FHMM is a statistical model that use latent variables in $\mathcal{X} = \{0, 1\}^{N \times K}$ to characterize time series data $y \in \mathbb{R}^N$. Denote $p(x)$ for hidden variables, and $p(y|x)$ for likelihood:

$$p(x) = \prod_{n=1}^{N} p(x_{n,1}) \prod_{k=2}^{K} p(x_{n,k}|x_{n,k-1}), \quad p(y|x) = \prod_{n=1}^{N} \text{Gaussian}(y_n; wx_n + b, \sigma^2) \tag{11}$$

Given data $y$, we use MCMC to sample $x$ from the posterior $p(x|y)$ and compare the mixing for different samplers. We choose parameters $N = 1000, K = 10, \mathbb{P}(x_{n,1} = 1) = 0.05$,

---
[*]Computed follows Tensorflow Probability

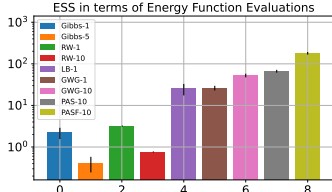 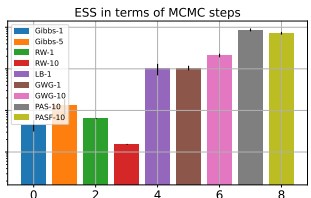 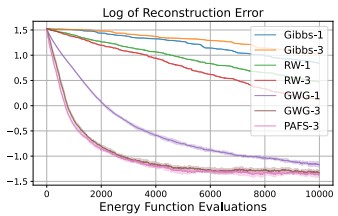

Figure 4: ESS for Different Samplers on $200 \times 200$ Ising model

Figure 5: BurnIn in FHMM

$\mathbb{P}(X_{n,k} = x_{n,k-1}) = 0.85$, $w \in$ Gaussian$(0, I_K)$, $b \in$ Gaussian$(0, 1)$, and $\sigma^2 = 0.25$. We run each chain 5 times and report the mean and std for the energy (negative log joint density) $E(\hat{x}) = -\log p(\hat{x})p(y|\hat{x})$ and the reconstruction error $\|y - w\hat{x}\|_2$ w.r.t. the number of evaluations of energy function. From figure 5 we can see that both GWG and PAFS gain significant improvements in mixing by using a larger neighborhood. More results are given in Sec B.2

**Restricted Boltzmann Machine:** RBM is bipartite latent-variable model, whose energy function is defined as:

$$f(x) = \log(1 + e^{w^T x + c}) + b^T x \tag{12}$$

where $\{W, b, c\}$ are parameters and $x \in \{0, 1\}^D$. We follow Grathwohl et al. (2021) to train a RBM with 500 hidden units on the MNIST dataset using contrastive divergence(Hinton, 2002). We generate samples via various MCMC samplers on the trained RBM. Besides reporting ESS, we also estimate the maximum mean discrepancy (MMD) (Gretton et al., 2012) w.r.t. a set of "ground truth" samples generated by structure known Block-Gibbs sampler. Figure 7 shows PAFS has large ESS and can efficiently match the "ground truth" samples. We also demonstrate an ablation study on different the approximations of the auxiliary path. Specifically, we evaluate GWG, PAFS with different lengths. For each sampler, we run 100 chains and report their ESS and average hop distance. We can notice GWG suffers from a large path length $L$, while our PAFS obtains robust

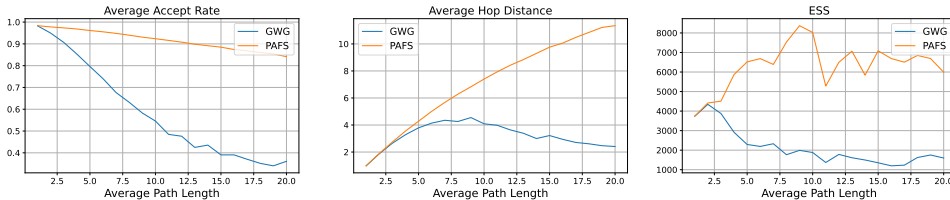

Figure 6: Sampling on RBM with Different Path Length

improvements for all $L$ by employing a soft no-replacement sampling. This result indicates that our PAFS provides an efficient framework to sample from a large neighborhood. More results and discussions, including a no-replacement sampler, are given in Sec B.3.

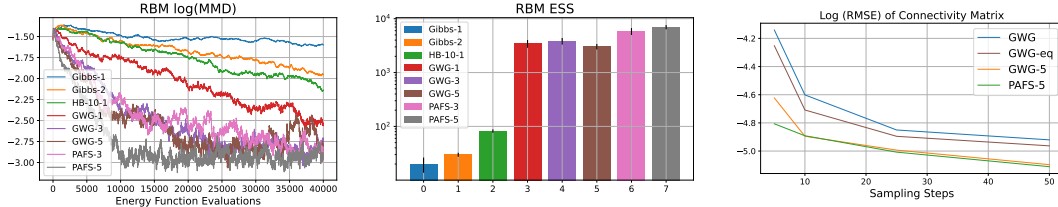

Figure 7: Sampling on RBM trained on MNIST

Figure 8: Learning Ising

| Dataset | VAE (MLP) | VAE (Conv) | EBM (GWG) | EBM (Gibbs) | EBM (PAFS) | RBM | DBN |
|---|---|---|---|---|---|---|---|
| Static MNIST | -86.05 | -82.41 | -80.01 | -117.17 | **-79.58** | -86.39 | -85.67 |
| Dynamic MNIST | -82.42 | -80.40 | -80.51 | -121.19 | **-79.59** | - | - |
| Omniglot | -103.52 | -97.65 | -94.72 | -142.06 | **-90.75** | -100.47 | -100.78 |
| Caltech Silhouettes | -112.08 | -106.35 | -96.20 | -163.50 | **-84.56** | - | - |

Table 1: Evaluation of different discrete models on the held-out test set.

## 5.4 LEARNING ON ENERGY BASED MODEL

Learning an EBM is a challenge task. Consider the target distribution is $\pi$ and our energy function $f_\theta$ is parameterized by $\theta$. The gradient for the likelihood of $\pi_\theta(x) \propto e^{-f_\theta(x)}$ is:

$$\nabla_\theta \log p(x) = \mathbb{E}_\pi[\nabla_\theta f_\theta(x)] - \mathbb{E}_{\pi_\theta}[\nabla_\theta f_\theta(x)] \tag{13}$$

The first expectation can be estimated using the data from true distribution. The second expectation requires samples from the current model, which is typically obtained via MCMC. Hence, the success of training an EBM relies on efficient MCMC algorithms to get an accurate estimation of the second expectation.

**Ising model** We first learn an Ising model following the setting in Grathwohl et al. (2021), where the energy function $f(x) = \theta x^T J x$. Given a set of samples $\{x_i\}$, the task is to learn an EBM via recovering the connectivity matrix $J$. We generate a $25 \times 25$ 2D cyclic lattice with $\theta = 0.25$ and sample training data with a long-run Gibbs chain. We train the models to maximize the likelihood of samples using persistent contrastive divergence (PCD)(Tieleman & Hinton, 2009) and report the root mean squared error (RMSE) between the inferred connectivity matrix $\hat{J}$ and the true matrix $J$. For fair comparison, we also include GWG-eq, which runs GWG-1 with more steps such that it has the same time cost as our PAFS-5. Figure 8 shows larger neighborhood can effectively help learn the Ising model. GWG-5 having similar performance as PAFS-5 as, using PCD, the learning efficiency is mainly determined by the mixing of the Markov chain.

**Deep EBM** We evaluate path auxiliary sampler by learning a deep EBM. The experiment follows the setting in (Grathwohl et al., 2021). We train deep EBMs paramterized by Residual Networks(He et al., 2016) on small binary image datasets using PCD(Tieleman & Hinton, 2009) with a replay buffer(Du & Mordatch, 2019). We compare our methods with Variational Autoencoders Kingma & Welling (2013), GWG (Grathwohl et al., 2021), RBM and a Deep Belief Network(Hinton, 2009). We estimate the likelihoods using Annealed Importance Sampling(Neal, 2001) and report the results in table.1. The results for VAE is taken from Tomczak & Welling (2018), RBM and DBN are taken from Burda et al. (2015), GWG is taken from Grathwohl et al. (2021). We can see that our approach improves the log-likelihoods for deep EBMs on all datasets.

## 6 DISCUSSION AND CONCLUSION

The problem for the selection of the path length remains open. In this work, we propose a soft no-replacement method PAFS. PAFS is not sensitive to the path length larger than the optimal length, hence has robust proposal quality for different choice of path length. We also considered no-backtrack samplers that flip sites without replacement. Intuitively, such auxiliary path encourages to propose states in larger distance. However, for no-backtrack samplers, a too large path length is harmful for the proposal quality and significant reduce the efficiency of the sampler. As a result, finding a principle way to decide the path length is very important. In continuous space, the optimal step size can be characterized via acceptance rate (Gelman et al., 1997; Roberts & Rosenthal, 1998; 2001; Beskos et al., 2013). PAS is the gradient based sampler in discrete space, we believe it is possible to derive its optimal path length in a similar manner as continuous case. We will investigate it in our future work.

In summary, informed proposal has shown good results for inference, sampling, and learning EBMs in discrete spaces. Our path auxiliary sampler provides an approach allowing informed proposal to efficiently explore large neighborhoods. We believe there is considerable room for future works to improve the sampling methods in discrete space.

**Acknowledgement**: This research was supported in part by the Defense Advanced Research Projects Agency (DARPA) under Contract FA8750-20-C-0542 (Systemic Generative Engineering). The views, opinions, and/or findings expressed are those of the author(s) and should not be interpreted as representing the official views or policies of the Department of Defense or the U.S. Government.

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

## A   PROOFS

### A.1   PROOF FOR THEOREM 1

Our path auxiliary sampler is a proposal distribution augmented auxiliary sampler. Before diving into the proof, we first introduce some background knowledge for auxiliary sampler, based on the results from page 86 in Liang et al. (2011). Given a proposal distribution $T(y|x)$, we can augment it by an auxiliary variable $u$, such that $T(y|x) = \int T_1(u|x)T_2(y|x, u)du$. Using this proposal distribution, we can define a marginal sampler and an auxiliary sampler.

1. For a marginal sampler, the accept rate is

$$A_{\mathrm{mar}}(x, y) = \min\{1, \frac{\pi(y)\int T_1(u|y)T_2(x|y, u)du}{\pi(x)\int T_1(u|x)T_2(y|x, u)du}\} = \min\{1, \frac{\pi(y)T(x|y)}{\pi(x)T(y|x)}\} \tag{14}$$

and for $y \neq x$, its transition kernel is:

$$A_{\mathrm{mar}}(x, y) = T(y|x)A_{\mathrm{mar}}(x, y) \tag{15}$$

2. For an auxiliary sampler, given $x$, we first sample auxiliary $u \sim T_1(u|x)$, then we sample the new state $y \sim T_2(y|x, u)$. The corresponding accept rate is:

$$A_{\mathrm{aux}}(x, y, u) = \min\{1, \frac{\pi(y)T_1(u|y)T_2(x|y, u)}{\pi(x)T_1(u|x)T_2(y|x, u)}\} \tag{16}$$

Since the proposal and the accept rate depend on the auxiliary $u$, we need to integrate the auxiliary variable $u$ to obtain the transition from $x$ to $y$:

$$K_{\mathrm{aux}}(x, y) = \int_u T_1(u|x)T_2(y|x, u)A(x, y, u)du \tag{17}$$

Though a marginal sampler is Peskun better than an auxiliary sampler (Titsias & Papaspiliopoulos, 2018), a marginal sampler is intractable in most of scenarios, as it requires to integrate over auxiliary variables $u$ in both proposal and accept rate calculation. Hence, our sampler is implemented in an auxiliary fashion.

Going back our theorem, the auxiliary variable $u$ is the auxiliary path $(\sigma, L)$. Condition on $L$, we have:

$$T_1(u|x, L) = T_1(\sigma|x) = \prod_{l=1}^{L} Q_0(\sigma_{l-1}, \sigma_l) \tag{18}$$

$$T_2(y|x, u, L) = \mathbb{I}\{x, y \text{ are the two ends of path } u(\text{or say}\sigma)\} \tag{19}$$

Given path length $L$, with a little abuse of notation, the transition kernel for our path auxiliary sampler $K(x, y|L)$ is:

$$\underbrace{\sum_{\substack{(\sigma, L)\in\Sigma(\mathcal{X}, \mathcal{N}): \\ \sigma_0 = x}} \left[ \overbrace{\left(\prod_{l=1}^{L} Q_0(\sigma_{l-1}, \sigma_l)\right)}^{T_1(\sigma|x, L)} \overbrace{\mathbb{I}\{\sigma_L = y\}}^{T_2(y|x, \sigma, L)} \overbrace{\min\left\{1, \frac{\pi(y)\prod_{l=1}^{L} Q_0(\sigma_l, \sigma_{l-1})\mathbb{I}\{\sigma_0 = x\}}{\pi(x)\prod_{l=1}^{L} Q_0(\sigma_{l-1}, \sigma_l)\mathbb{I}\{\sigma_L = y\}}\right\}}^{A(x, y, \sigma|L)} \right]}_{\substack{T_1(\sigma|x, L)T_2(y|x, \sigma, L)A(x, y, \sigma|L)d\sigma \\ K(x, y|L) = \int_\sigma T_1(\sigma|x, L)T_2(y|x, \sigma, L)A(x, y, \sigma|L)d\sigma}}$$

$$\tag{20}$$

When $\sigma_L \neq y$, the accept rate above is not well-defined. To avoid making extra definition, we absorb $T_2$, the indicator, as a constraint of the domain in the integration. Then, we have:

$$\underbrace{\sum_{\substack{(\sigma, L)\in\Sigma(\mathcal{X}, \mathcal{N}): \\ \sigma_0 = x, \sigma_L = y}} \left[ \overbrace{\left(\prod_{l=1}^{L} Q_0(\sigma_{l-1}, \sigma_l)\right)}^{T_1(\sigma|x, L)} \overbrace{\min\left\{1, \frac{\pi(y)\prod_{l=1}^{L} Q_0(\sigma_l, \sigma_{l-1})}{\pi(x)\prod_{l=1}^{L} Q_0(\sigma_{l-1}, \sigma_l)}\right\}}^{A(x, y, \sigma|L)} \right]}_{K(x, y|L) = \int_\sigma T_1(\sigma|x, L)T_2(y|x, \sigma, L)A(x, y, \sigma|L)d\sigma} \tag{21}$$

With these knowledge prepared, we begin to proof our theorem 1.

*Proof.* Denote $K(x, y)$ as the probability that $x$ transit to a different state $y$, then we have:

$$\pi(x)K(x, y) \tag{22}$$

$$= \pi(x) \sum_L \alpha(L) \sum_{\substack{(\sigma, L) \in \Sigma(\mathcal{X}, \mathcal{N}): \\ \sigma_0 = x, \sigma_L = y}} \left[ \left( \prod_{l=1}^L Q_0(\sigma_{l-1}, \sigma_l) \right) \min \left\{ 1, \frac{\pi(y) \prod_{l=1}^L Q_0(\sigma_l, \sigma_{l-1})}{\pi(x) \prod_{l=1}^L Q_0(\sigma_{l-1}, \sigma_l)} \right\} \right] \tag{23}$$

$$= \sum_L \alpha(L) \sum_{\substack{(\sigma, L) \in \Sigma(\mathcal{X}, \mathcal{N}): \\ \sigma_0 = x, \sigma_L = y}} \left[ \pi(x) \left( \prod_{l=1}^L Q_0(\sigma_{l-1}, \sigma_l) \right) \min \left\{ 1, \frac{\pi(y) \prod_{l=1}^L Q_0(\sigma_l, \sigma_{l-1})}{\pi(x) \prod_{l=1}^L Q_0(\sigma_{l-1}, \sigma_l)} \right\} \right] \tag{24}$$

$$= \sum_L \alpha(L) \sum_{\substack{(\sigma, L) \in \Sigma(\mathcal{X}, \mathcal{N}): \\ \sigma_0 = x, \sigma_L = y}} \min \left\{ \pi(x) \prod_{l=1}^L Q_0(\sigma_{l-1}, \sigma_l), \pi(y) \prod_{l=1}^L Q_0(\sigma_l, \sigma_{l-1}) \right\} \tag{25}$$

$$= \sum_L \alpha(L) \sum_{\substack{(\sigma, L) \in \Sigma(\mathcal{X}, \mathcal{N}): \\ \sigma_0 = x, \sigma_L = y}} \left[ \pi(y) \left( \prod_{l=1}^L Q_0(\sigma_l, \sigma_{l-1}) \right) \min \left\{ \frac{\pi(x) \prod_{l=1}^L Q_0(\sigma_{l-1}, \sigma_l)}{\pi(y) \prod_{l=1}^L Q_0(\sigma_l, \sigma_{l-1})}, 1 \right\} \right] \tag{26}$$

$$= \pi(y) \sum_L \alpha(L) \sum_{\substack{(\sigma, L) \in \Sigma(\mathcal{X}, \mathcal{N}): \\ \sigma_0 = y, \sigma_L = x}} \left[ \left( \prod_{l=1}^L Q_0(\sigma_l, \sigma_{l-1}) \right) \min \left\{ \frac{\pi(x) \prod_{l=1}^L Q_0(\sigma_l, \sigma_{l-1})}{\pi(y) \prod_{l=1}^L Q_0(\sigma_{l-1}, \sigma_l)}, 1 \right\} \right] \tag{27}$$

$$= \pi(y)K(y, x) \tag{28}$$

The key idea for the proof is that $(\sigma, L)$ is symmetric w.r.t. its two ends, when $\sigma$ is a path from $x$ to $y$, $\sigma$ is also a path from $y$ to $\sigma$. Hence, we are able to exchange the orientation of the path in equation 27. $\qquad \square$

## A.2 PROOF FOR LEMMA 1

**Lemma 1.** *$\mathcal{X}$ is a finite state space with distribution $\pi$ and a neighborhood function $\mathcal{N}$ such that every state has equal number of states. $\mathcal{F}$ is a class of weight functions, such that for any $f \in \mathcal{F}$, 1) $f : \mathbb{R}_+ \to \mathbb{R}_+$, 2) $f(1) = 1$, 3) $f(t)$ is monotonically increasing, 4) $f(t)f(\frac{1}{t})t \leq 1, \forall t \leq 1$. $\forall f \in \mathcal{F}$, define $\bar{f}(t) = \sqrt{f(t)f(\frac{1}{t})t}$, we have*

$$\min_{x \in \mathcal{X}} Z_{\bar{f}}(x) \leq \max_{y \in \mathcal{X}} Z_f(y) \tag{29}$$

*Proof.* Since $\mathcal{X}$ is finite, we can always find state $x_1$, $x_2$ having highest and lowest probability, respectively. Then, we have:

$$Z_{\bar{f}}(x_1) = \sum_{z \in \mathcal{N}(x_1)} \bar{f}\left(\frac{\pi(z)}{\pi(x_1)}\right) \leq |\mathcal{N}(x_1)| = |\mathcal{N}(x_2)| \leq \sum_{z \in \mathcal{N}(x_2)} f\left(\frac{\pi(z)}{\pi(x_2)}\right) = Z_f(x_2) \tag{30}$$

$\qquad \square$

## A.3 DISCUSSION FOR IDEAL FUNCTION

We define ideal function as a stepping stone to show the advantage of locally balanced function. We name it "ideal" as the its properties are ideal assumption for a weight function in PIP. Conditions 1) $f : \mathbb{R}_+ \to \mathbb{R}_+$ is a natural requirement for weight function. Condition 2) $f(1) = 1$ can be easily realized by substituting $f(t)$ by $f(t)/f(1)$, as PIP only depends on the ratio of weights. condition 3) $f(t)$ is monotonically increasing, and condition 4) $f(t)f(\frac{1}{t})t \leq 1, \forall t \leq 1$, though, are technical requirements, it is reasonable for weight functions. Monotonic increasing in 3) indicates the proposal distribution match the target distribution, where a point $z$ has a higher probability in target distribution should also has a higher probability in proposal distribution. Condition 4) is weaker than the following condition: $f(t) \leq t, \forall t \geq 1$, hence condition 4) is easier to satisfy.

Currently definition of ideal function class already contains most of commonly used weight function, such as $f(t) = \frac{2t}{1+t}$, $f(t) = \max\{1, t\}$, $f(t) = \min\{1, t\}$, and $f(t) = t^\alpha$, where $\alpha \geq 0$. Also, we can following properties

**Property 2.** *The logarithm of ideal function is closed in convex combination. That's to say,* $\forall f_1, ..., f_n \in \mathcal{F}$, *and* $\forall \lambda_1, ..., \lambda_n$, *s.t* $\lambda_i \geq 0$ *and* $\sum_{i=1}^n \lambda_i = 1$, *we have*

$$F(t) = e^{\sum_{i=1}^n \lambda_i \log f_i(t)} \in \mathcal{F} \tag{31}$$

*Proof.* Let $f_i$ and $\lambda_i$ be defined as above. Obviously, $F$ is positive and $F(1) = 1$, hence satisfies condition 1) and 2). For condition 3), we have:

$$F(t)F(\frac{1}{t})t = e^{\sum_{i=1}^n \lambda_i \log f_i(t)} \cdot e^{\sum_{i=1}^n \lambda_i \log f_i(\frac{1}{t})} \cdot e^{\sum_{i=1}^n \lambda_i \log t} \tag{32}$$

$$= e^{\sum_{i=1}^n \lambda_i (\log f_i(t) + \log f_i(\frac{1}{t}) + \log t)} \tag{33}$$

$$= e^{\sum_{i=1}^n \lambda_i \log(f_i(t) f_i(\frac{1}{t})t)} \tag{34}$$

Hence, $\forall t \leq 1$, we have:

$$F(t)f(\frac{1}{t})t \leq e^{\sum_{i=1}^n \lambda_i 0} = 1 \tag{35}$$

which implies $F(t) \in \mathcal{F}$. $\qquad\square$

**Property 3.** *Ideal function is a superset for normalized monotonically increasing locally balanced function* $\mathcal{G}_I := \{g \in \mathcal{G} : g(1) = 1, g \text{ is monotonically increasing}\}$.

*Proof.* $\forall g \in \mathcal{G}_I$, condition 1), 2) and 3) are obviously satisfied. For condition 4), since $g(t)$ is locally balanced, we have:

$$g(t)g(\frac{1}{t})t = g^2(t) \leq g^2(1) = 1 \tag{36}$$

implies $g \in \mathcal{F}$. $\qquad\square$

Considering the ideal function cover such a large number of functions, we believe the asymptotic optimality over ideal function strongly suggests locally balanced function is a good choice for path auxiliary sampler.

## A.4 PROOF FOR LEMMA 2

The (undirected) conditional independence graph corresponds to Markov random fields:

**Definition 2.** *The conditional independence graph of* $X$ *is the undirected graph* $G = (K, E)$, *where* $K = \{1, ..., k\}$ *and* $(i, j) \notin E$ *if and only if* $X_i \perp X_j | X_{K \setminus \{i,j\}}$.

More details can be found in p.60 Whittaker (2009).

**Lemma 2.** *Consider the state space in Cartesian products* $\mathcal{X} = \times_{i=1}^n \mathcal{X}_i$, *where each* $\mathcal{X}_i$ *is a finite space with* $M$ *elements, and the neighborhood is defined as 1-Hamming ball. Let* $d_n$ *be the maximum degree in conditional independence graph. Define* $c_g^{(n)} := \sup_{y \in \mathcal{N}(x)} \frac{Z_g(y)}{Z_g(x)}$. *If 1)* $\lim_{n \to \infty} \frac{d_n}{n} = 0$; *2) the target distribution satisfies* $\frac{\pi(y)}{\pi(x)} \leq C < \infty, \forall y \in \mathcal{N}(x)$, *then we have:*

$$1 \leq c_g^{(n)} \leq 1 + \mathcal{O}(\frac{d_n g(C)}{n g(\frac{1}{C})}) \quad as \ n \to \infty, \quad \forall g \in \mathcal{F} \tag{37}$$

To simplify the notation, we assume $M = 2$ in this proof. It is straightforward to extend the proof to any finite $M$.

*Proof.* On one side, by definition, we can easily see that $c_g^{(n)} \geq 1$. On the other side, given $x$, we use index $i$ represent that we select $y \in \mathcal{N}(x)$ by flipping index $i$ for $x$. We denote $g_x(j) = g(\pi(z))/g(\pi(x))$ where $z$ is obtained by flipping index $j$ for $x$, and we denote $g_i(j) = g(\pi(z))/g(\pi(y))$, where $z$ is obtained by flipping index $j$ for $y$. We also denote $B(i)$

as the Markov boundary for $i$, which means given $B(i)$, $i$ is independent with remaining nodes. Then, we can write

$$c_g^{(n)} = \sup_{x \in \mathcal{X}} \sup_{i \in [M]} \frac{\sum_{j=1}^n g_i(j)}{\sum_{j=1}^n g_x(j)} \tag{38}$$

$$= \sup_{x \in \mathcal{X}} \sup_{i \in [M]} \frac{\sum_{j \in B(i)} g_i(j) + \sum_{j \notin B(i)} g_i(j)}{\sum_{j \in B(i)} g_x(j) + \sum_{j \notin B(i)} g_x(j)} \tag{39}$$

$$\leq \sup_{x \in \mathcal{X}} \sup_{i \in [M]} \frac{\sum_{j \in B(i)} g_i(j)}{\sum_{j \in B(i)} g_x(j)} + \frac{\sum_{j \notin B(i)} g_i(j)}{\sum_{j \in B(i)} g_x(j) + \sum_{j \notin B(i)} g_x(j)} \tag{40}$$

$$\leq \sup_{x \in \mathcal{X}} \sup_{i \in [M]} 1 + \frac{\sum_{j \notin B(i)} g_i(j)}{\sum_{j \in B(i)} g_x(j) + \sum_{j \notin B(i)} g_x(j)} \tag{41}$$

$$\leq \sup_{x \in \mathcal{X}} \sup_{i \in [M]} 1 + \frac{d_n g(C)}{n g(\frac{1}{C})} \tag{42}$$

$$\leq 1 + \frac{d_n g(C)}{n g(\frac{1}{C})} \tag{43}$$

If $B(i)$ is not empty, the first term in equation 40 equals to 1. If $B(i)$ is empty, the first term in equation 40 does not exist, hence it can still be bounded by 1. $\qquad\square$

**Remark**: For a fixed ideal function $g \in \mathcal{F}$, Lemma 2 shows $c^{(n)}$ converges to 1 at a rate $1 + \mathcal{O}(\frac{d_n}{n})$

### A.5 PROOF FOR THEOREM 2

We prove the theorem by using Lemma 1 and Lemma 2.

*Proof.* We first show that, for all $g \in \mathcal{F}$, the transition probability:

$$P_{\bar{g}}(x, y) \geq (\frac{1}{c_g c_{\hat{g}}})^U P_g(x, y) \tag{44}$$

where $c_g$ is defined in Lemma 2. We temporarily ignore the superscription and will add it back at the end of the proof. Consider a path $(\sigma, L)$, we denote $t_{j,k} = \frac{\pi(\sigma_k)}{\pi(\sigma_j)}$. Then the probability that $x = \sigma_0$ transits to $y = \sigma_L$ is:

$$P_g(x, y, \sigma | L) = \prod_{l=1}^L Q_g(\sigma_{l-1}, \sigma_l) \min \left\{ 1, \frac{\pi(y) \prod_{l=1}^L Q_g(\sigma_l, \sigma_{l-1})}{\pi(x) \prod_{l=1}^L Q_g(\sigma_{l-1}, \sigma_l)} \right\} \tag{45}$$

$$= \min \left\{ \prod_{l=1}^L \frac{g(t_{l-1,l})}{Z_g(\sigma_{l-1})}, \prod_{l=1}^L t_{l-1,l} \frac{g(t_{l,l-1})}{Z_g(\sigma_l)} \right\} \tag{46}$$

By definition of $c_g$, we have:

$$P_g(x, y, \sigma | L) \leq c_g^L \frac{\min \left\{ \prod_{l=1}^L g(t_{l-1,l}), \prod_{l=1}^L t_{l-1,l} g(t_{l,l-1}) \right\}}{\max_{z \in \mathcal{X}^{(n)}} Z_g^L(z)} \tag{47}$$

$$P_{\bar{g}}(x, y, \sigma | L) \geq \frac{1}{c_{\bar{g}}^L} \frac{\min \left\{ \prod_{l=1}^L \bar{g}(t_{l-1,l}), \prod_{l=1}^L t_{l-1,l} \bar{g}(t_{l,l-1}) \right\}}{\min_{z \in \mathcal{X}^{(n)}} Z_{\bar{g}}^L(z)} \tag{48}$$

Use the property of $\bar{g}$, we have

$$\min \left\{ \prod_{l=1}^L g(t_{l-1,l}), \prod_{l=1}^L t_{l-1,l} g(t_{l,l-1}) \right\} \leq \sqrt{\prod_{l=1}^L g(t_{l-1,l}) g(t_{l,l-1}) t_{l-1,l}} = \prod_{l=1}^L \bar{g}(t_{l-1,l}) \tag{49}$$

Combining equation 47, equation 48, equation 49, and Lemma 1 we have:

$$P_{\bar{g}}(x,y,\sigma|L) \geq \frac{1}{c_{\bar{g}}^L} \frac{\prod_{l=1}^L \bar{g}(t_{l-1,l})}{\min_{z \in \mathcal{X}^{(n)}} Z_{\bar{g}}^L(z)} \tag{50}$$

$$\geq \frac{1}{c_{\bar{g}}^L} \frac{\min\left\{\prod_{l=1}^L g(t_{l-1,l}), \prod_{l=1}^L t_{l-1,l}g(t_{l,l-1})\right\}}{\max_{z \in \mathcal{X}^{(n)}} Z_g^L(z)} \tag{51}$$

$$\geq \frac{1}{c_{\bar{g}}^L c_g^L} P_g(x,y,\sigma|L) \tag{52}$$

The inequality holds for arbitrary auxiliary path $(\sigma, L)$ and the path length $L \leq U$, hence we prove the first step. Then $\forall g \in \mathcal{F}$, we use the estimation of $c_g^{(n)}$ in Lemma 2, we have

$$P_{\bar{g}}(x,y) \geq C_g^{(n)} P_g(x,y) \tag{53}$$

where

$$C_g^{(n)} = 1 - \mathcal{O}(U \frac{d_n g(C)}{n g(\frac{1}{C})}) \tag{54}$$

This indicates $\bar{g}$ is asymptotically better than $g$ and proves the theorem. □

**Remark**: In the proof, equation 54 shows the convergence rate depends on $g$, this factor prevents a uniform convergence rate for the ideal function class $\mathcal{F}$. We can obtain a uniform rate when we constraint in smaller function class, for example, if we restrict to $\mathcal{F}_M := \{f \in \mathcal{F} : f(C)/f(\frac{1}{C}) \leq M\}$, the convergence rate is $1 - \mathcal{O}(\frac{d_n}{n})$.

### A.6 PROOF FOR PROPERTY 1

*Proof.*

$$A(x,y,\sigma,L) = \min\left\{1, \frac{\pi(y)\prod_{l=1}^L Q(\sigma_l, \sigma_{l-1})}{\pi(x)\prod_{l=1}^L Q(\sigma_{l-1}, \sigma_l)}\right\} \tag{55}$$

$$= \min\left\{1, \left(\prod_{l=1}^L \frac{\pi(\sigma_l)}{\pi(\sigma_{l-1})}\right) \frac{\prod_{l=1}^L g(\frac{\pi(\sigma_{l-1})}{\pi(\sigma_l)})/Z_g(\sigma_l)}{\prod_{l=1}^L g(\frac{\pi(\sigma_l)}{\pi(\sigma_{l-1})})/Z_g(\sigma_{l-1})}\right\} \tag{56}$$

$$= \min\left\{1, \left(\prod_{l=1}^L \frac{\pi(\sigma_l)}{\pi(\sigma_{l-1})} \frac{g(\frac{\pi(\sigma_{l-1})}{\pi(\sigma_l)})}{g(\frac{\pi(\sigma_l)}{\pi(\sigma_{l-1})})}\right) \left(\prod_{l=1}^L \frac{Z_g(\sigma_{l-1})}{Z_g(\sigma_l)}\right)\right\} \tag{57}$$

$$= \min\left\{1, \frac{Z_g(x)}{Z_g(y)}\right\} \tag{58}$$

□

### A.7 PROOF FOR THEOREM 3

The idea to proof theorem 3 is similar to Grathwohl et al. (2021), which uses the Lipschitz condition to bound the estimation error. The different is, in PAFS, we use the linearization at each state $\sigma_l$ to propose the next state $\sigma_{l+1}$, hence to need to accumulate the error at each step to obtain our final result.

*Proof.* For any path $(\sigma, L)$, we first bound the estimation error $\tilde{f}_0(z) - \tilde{f}_0(\sigma_{l-1})$, for $z \in \mathcal{N}(\sigma_{l-1})$. Since $f$ is $M$-smooth, we have:

$$f(z) - f(\sigma_{l-1}) \tag{59}$$

$$\leq \langle \nabla f(\sigma_{l-1}), z - \sigma_{l-1} \rangle + \frac{K}{2} \|z - \sigma_{l-1}\|^2 \tag{60}$$

$$= \langle \nabla f(\sigma_0), z - \sigma_{l-1} \rangle + \langle \nabla f(\sigma_{l-1}) - \nabla f(\sigma_0), z - \sigma_{l-1} \rangle + \frac{K}{2} \|z - \sigma_{l-1}\|^2 \tag{61}$$

$$\leq \langle \nabla f(\sigma_0), z - \sigma_{l-1} \rangle + K(l - \frac{1}{2}) \tag{62}$$

Similarly, we also have:

$$f(z) - f(\sigma_{l-1}) \geq \langle \nabla f(\sigma_0), z - \sigma_{l-1} \rangle + K(l + \frac{1}{2}) \tag{63}$$

With this estimation, we have:

$$\tilde{Q}_0(\sigma_{l-1}, \sigma_l) = \frac{e^{-\frac{1}{2} \langle \nabla f(\sigma_0), \sigma_l - \sigma_{l-1} \rangle}}{\sum_{z \in \mathcal{N}(\sigma_{l-1})} e^{-\frac{1}{2} \langle \nabla f(\sigma_0), \sigma_l - \sigma_{l-1} \rangle}} \tag{64}$$

$$\geq \frac{e^{-\frac{1}{2}[f(\sigma_l) - f(\sigma_{l-1}) - K(l - \frac{1}{2})]}}{\sum_{z \in \mathcal{N}(\sigma_{l-1})} e^{-\frac{1}{2}[f(z) - f(\sigma_{l-1}) + K(l + \frac{1}{2})]}} \tag{65}$$

$$= Q(\sigma_{l-1}, \sigma_l) e^{-Kl} \tag{66}$$

Similarly, we can also obtain

$$\tilde{Q}_L(\sigma_l, \sigma_{l-1}) \geq Q(\sigma_l, \sigma_{l-1}) e^{K(L-l)} \tag{67}$$

Now, consider the transition kernel

$$\tilde{P}(x, y) = \sum_L \alpha(L) \sum_{\substack{(\sigma, L) \in \Sigma(\mathcal{X}, \mathcal{N}): \\ \sigma_0 = x, \sigma_L = y}} \prod_{l=1}^{L} \tilde{Q}_0(\sigma_{l-1}, \sigma_l) \min \left\{ 1, \frac{\pi(y)}{\pi(x)} \frac{\prod_{l=1}^{L} \tilde{Q}_L(\sigma_l, \sigma_{l-1})}{\prod_{l=1}^{L} \tilde{Q}_0(\sigma_{l-1}, \sigma_l)} \right\} \tag{68}$$

$$= \sum_L \alpha(L) \sum_{\substack{(\sigma, L) \in \Sigma(\mathcal{X}, \mathcal{N}): \\ \sigma_0 = x, \sigma_L = y}} \min \left\{ \prod_{l=1}^{L} \tilde{Q}_0(\sigma_{l-1}, \sigma_l), \frac{\pi(y)}{\pi(x)} \prod_{l=1}^{L} \tilde{Q}_L(\sigma_l, \sigma_{l-1}) \right\} \tag{69}$$

$$\geq \sum_L \alpha(L) \sum_{\substack{(\sigma, L) \in \Sigma(\mathcal{X}, \mathcal{N}): \\ \sigma_0 = x, \sigma_L = y}} \min \left\{ \prod_{l=1}^{L} Q(\sigma_{l-1}, \sigma_l) e^{-Kl}, \frac{\pi(y)}{\pi(x)} \prod_{l=1}^{L} Q(\sigma_l, \sigma_{l-1}) e^{-K(L-l)} \right\} \tag{70}$$

$$\geq \sum_L \alpha(L) \sum_{\substack{(\sigma, L) \in \Sigma(\mathcal{X}, \mathcal{N}): \\ \sigma_0 = x, \sigma_L = y}} e^{-K \frac{L(L+1)}{2}} \min \left\{ \prod_{l=1}^{L} Q(\sigma_{l-1}, \sigma_l), \frac{\pi(y)}{\pi(x)} \prod_{l=1}^{L} Q(\sigma_l, \sigma_{l-1}) \right\} \tag{71}$$

$$\geq e^{-K \frac{U(U+1)}{2}} P(x, y) \tag{72}$$

$$\square$$

**Remark**: Though the theorem proved above is very loose, it provides a framework to prove approximation bound for path auxiliary sampler via single step approximation in equation 66. More sophisticated bounds can be obtained by improving the single step estimation in equation 66 via the property of the conditional independence. Specifically, a local modifying of the state usually does not influence most of the indices, which for example can be characterized by the Markov boundary. Then, with high probability, the auxiliary path can avoid manipulating correlated indices and hence result in the approximation error in equation 66 be controlled by a constant, rather than $e^{-Kl}$. For example, in Ising model, an auxiliary path has high probability to avoid manipulating two adjacent nodes, and in this case, the linearization is accurate and we loose nothing in approximation. In deep EBMs, the conditional independence is usually not available, but we can still expect low correlation between most of dimensions. Such analysis relies on more accurate assumption of conditional independence and we will leave it to our future work.

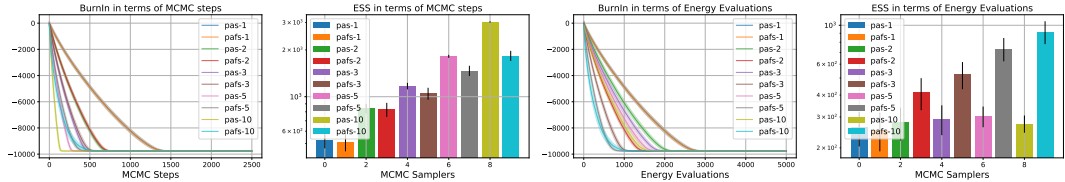

Figure 9: PAS and PAFS in different lengths on $50 \times 50$ Ising Model

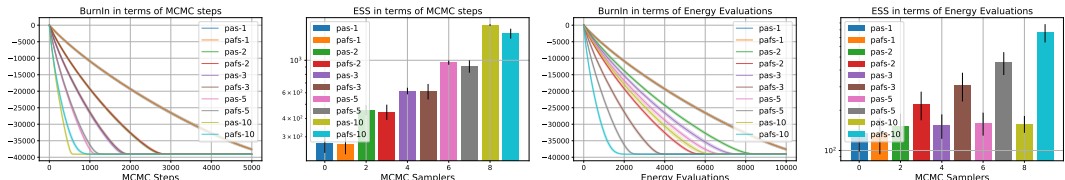

Figure 10: PAS and PAFS in different lengths on $100 \times 100$ Ising Model

# B  DETAILS FOR EXPERIMENTS

## B.1  SAMPLING ON ISING

Following Zanella (2020), we set the interaction term $\lambda = 1$, and we set $\alpha_i = \mu + Z_i$ if $i$ in the center of the square lattice, and $\alpha_i = -\mu + Z_i$ otherwise, where $\mu = 2$ and $Z_i \sim \text{Unif}(-3, 3)$. We generate the lattice Ising model in four sizes: $p = 50, 100, 150, 200$. We first run PAS and PAFS with path length $L = 1, 2, 3, 5, 10$ on Ising model with size $p = 50, 100, 150, 200$. For each path length and each model size, we run 100 chains 100,000 steps and compute the ESS using the last 50,000 steps. The results for $p = 200$ is given in figure 3. We give the results for $p = 50, 100, 150$ in the following. From figure 9, figure 10, figure 11, and figure 3, we can see that the difference between PAS and PAFS in terms of MCMC steps is decreasing when the model size is increasing. The reason is that when the model becomes larger, most of the nodes are conditionally independent. When the auxiliary path does not involve two adjacent nodes, the approximation has zero error. This observation provides a possibility to further improve the approximation bound in Theorem 3. We will study this in our future work.

For comparing with other samplers, we report the results for each problem size $p = 50, 100, 150, 200$ in figure 12, figure 13, figure 14, figure 4. Our path auxiliary samplers substantially outperform other competitors.

## B.2  SAMPLING ON FHMM

We compare PAS and PAFS with different path length on FHMM. We choose parameters $K = 10$, $\mathbb{P}(x_{n,1} = 1) = 0.05$, $\mathbb{P}(X_{n,k} = x_{n,k-1}) = 0.85$, $w \in \text{Gaussian}(0, I_K)$, $b \in \text{Gaussian}(0, 1)$, and $\sigma^2 = 0.25$. We simulate PAS, PAFS with path length $L = 1, 2, 3, 5, 10$ on FHMM with $N = 500, 1000, 1500, 2000$. For each configuration, we run 100 chains and report the burn in period, as well as the ESS in terms of MCMC steps and energy function evaluations.

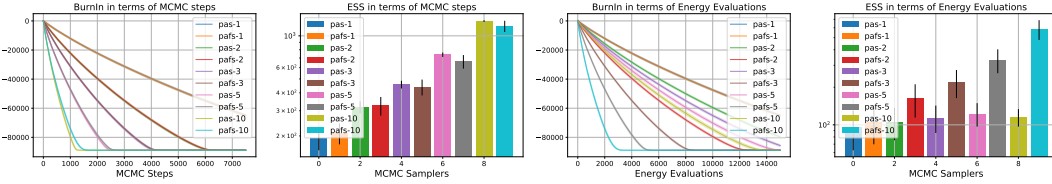

Figure 11: PAS and PAFS in different lengths on $150 \times 150$ Ising Model

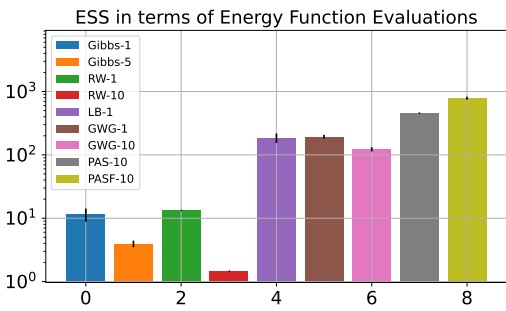 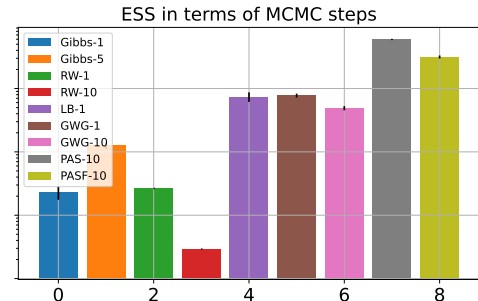

Figure 12: Sampling on $50 \times 50$ Ising model

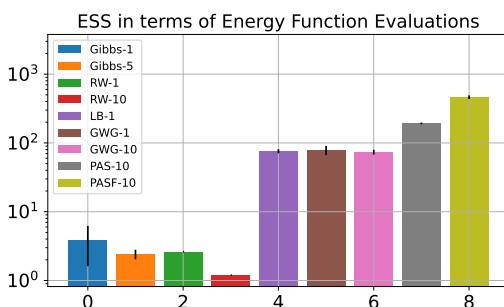 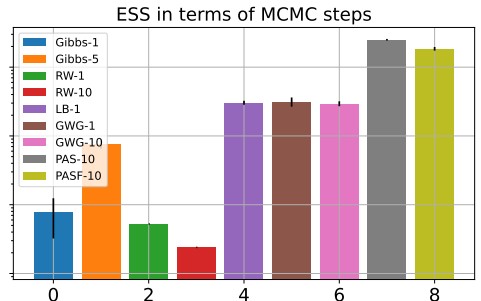

Figure 13: Sampling on $100 \times 100$ Ising model

We can observe that: 1) in terms of MCMC steps, PAFS has very similar burn in time in energy compared to PAS, while the the gap in ESS is larger than the gap in Ising. The reason is FHMM is not as close to linear model as Ising and PAFS has larger estimation error. 2) in terms of energy function evaluations, PAFS still leads the performance as other models. PAS only obtains faster mixing, and no improvements in ESS. The results show that whether PAS can help depends on the property of the target distribution.

### B.3 SAMPLING ON RBM

We compare GWG, PAFS, and NB with different path lengths on RBM trained on MNIST dataset. GWG (Grathwohl et al., 2021) samples all indices to manipulate based on the linearization at starting state of the path $\sigma_0$, and we classify it as sampling with replacement. Our PAFS samples the indices based on the linearization at current state $\sigma_l$ and we classify it as sampling with soft no replacement. Specifically, consider a binary case, when we flip index $i$ at state $\sigma_l$, then in the remaining of the path, the energy change for index $i$ will be reversed. Hence, if we flip an index and reduce the system's energy, we will have small probability flip it back. A third type sampler is No Backtrack (NB) sampler. It is also a path auxiliary sampler and it rules out the indices have been selected to assure

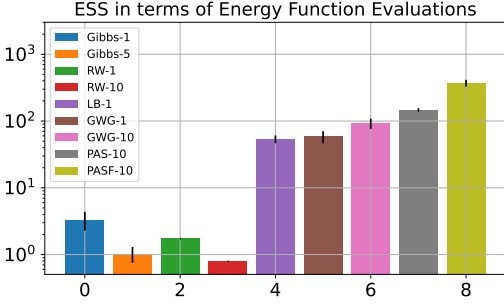 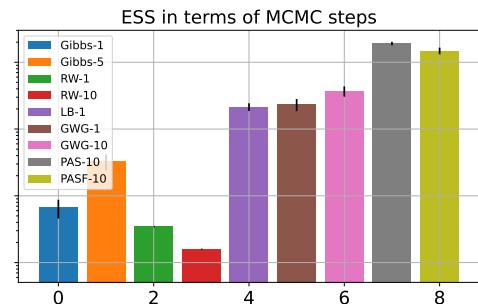

Figure 14: Sampling on $150 \times 150$ Ising model

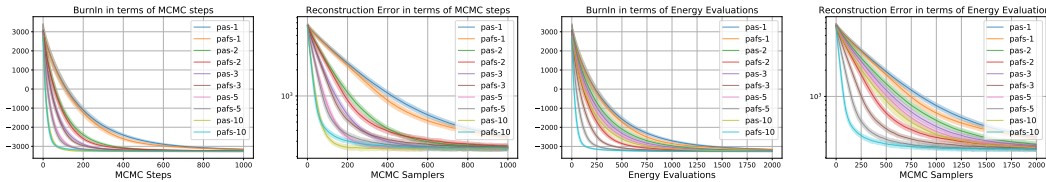

Figure 15: PAS and PAFS in different path lengths on FHMM N=500

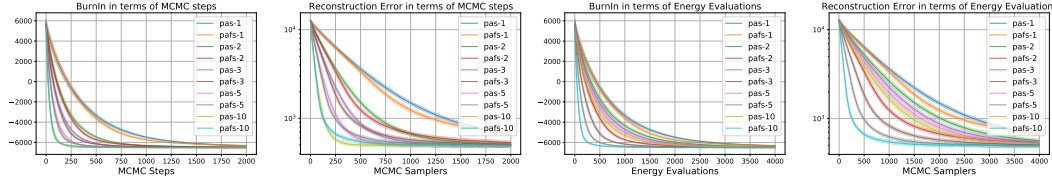

Figure 16: PAS and PAFS in different path lengths on FHMM N=1000

it is sampling with hard no replacement. Specifically, NB first calculate the probability for indices based on $\nabla f(\sigma_0)$ as GWG. Then, once selecting an index $i$ at $\sigma_l$, NB manipulates the probability for selecting index $i$ in the remaining path. For each sampler with path length $L = 1, ..., 20$, we run 100 chains and report the MMD (Gretton et al., 2012) w.r.t. a set of "ground truth" samples generated by structure known Block-Gibbs sampler. We can see that, when increasing the path length, PAFS has steady fast mixing, GWG decreases the efficiency in mixing, and NB fails in mixing. For this reason, we only compare the statistics for GWG and PAFS in our main text in figure 6.

## B.4 RUNNING TIME IN LEARNING ISING

We follow the experiments in Grathwohl et al. (2021). GWG runs faster 1.4x than our PAFS-5 as the energy function computation is cheap in this experiment. To obtain a fair comparison, we add GWG-equivalent-length (GWG-eq), which we allow GWG-1 to run 1.4x more sampling steps such that its running time is the same as our PAFS. When our PAFS runs 5, 10, 25, 50 steps, GWG-eq runs 6, 13 34, 69 steps. Hence, when we draw figure 8, the point for GWG-eq at step 5, 10, 25, 50 using the RMSE obtained at setp 6, 13, 34, 69.

## B.5 DETAILS IN LEARNING DEEP EBMS

| DataSet | Static MNIST | Omniglot | Caltech |
|---------|--------------|----------|---------|
| PAFS-3  | 0.685        | 0.885    | 0.893   |
| PAFS-5  | 0.746        | 0.980    | 0.962   |
| PAFS-7  | 0.775        | 1.030    | 1.010   |
| GWG     | 0.877        | 1.130    | 1.140   |

Table 2: Run Time for One Batch for different discrete models.

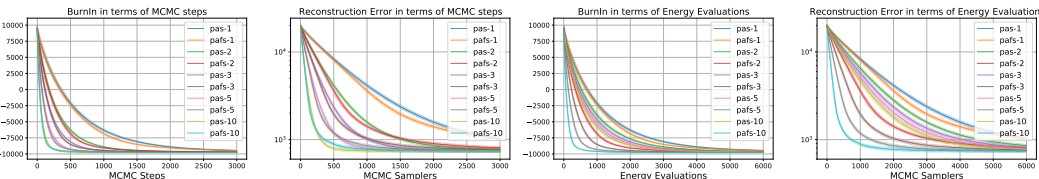

Figure 17: PAS and PAFS in different path lengths on FHMM N=1500

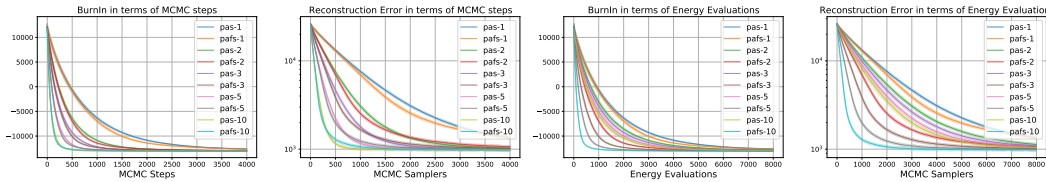

Figure 18: PAS and PAFS in different path lengths on FHMM N=2000

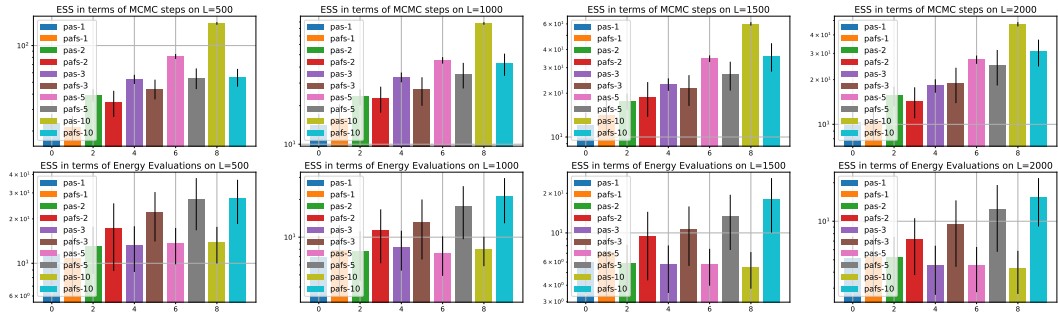

Figure 19: ESS for PAS and PAFS in different path lengths on FHMM

We use the same setting as Grathwohl et al. (2021), including the batch size, number of iterations, the PCD hyper parameters, etc. In principle, both PAFS and GWG requires two energy function evaluations per MCMC step, despite that PAFS explores much larger neighborhood space via path auxiliary. As the energy function evaluation and gradient calculation usually dominates the computation, we expect these two should have similar runtime.

In practice, we report the average running time per batch of 100 chains in table 2. We can see for deep EBMs, our PAFS with different path lengths run slightly faster than GWG. Given that both of the methods implemented using `pytorch`, the minor speed up might be due to the implementation issue. So overall we think it is fair to say the two methods run equally fast in most cases. For this reason, we use the same number of steps when we train the deep EBMs. As is reported in Grathwohl et al. (2021), we the same number of steps (40) to train the binary deep EBMs. For our method, we also tune the expected path length in $\{3, 5, 7\}$, and report the result with the best validation likelihood.

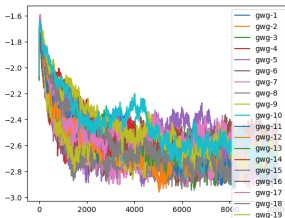

Figure 20: GWG Burn In

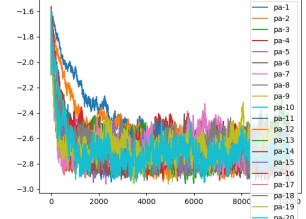

Figure 21: PAFS Burn In

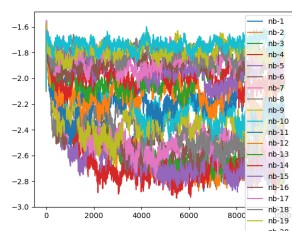

Figure 22: NB Burn In

