# OpenReview forum: "Path Auxiliary Proposal for MCMC in Discrete Space"
_ICLR.cc/2022/Conference — ICLR 2022 Spotlight_

### Official Review · Reviewer_CSos · 2021-10-23

**Correctness:** 4
**Technical Novelty And Significance:** 3
**Empirical Novelty And Significance:** 3
**Recommendation:** 8
**Confidence:** 5

**Details Of Ethics Concerns:**

None.

**Main Review:**

Strengths:

This paper presents a simple improvement over LB and GWG which alleviates the largest issue with these two samplers -- their inability to escape local energy wells. For LB/GWG to escape such a well we would need a long series of *accepted* transitions through low-likelihood regions, meaning we need the proposal to choose these unlikely transitions *and* the A/R step to accept them. Conversely, for PAS-X we only need to choose these unlikely transitions and the accept will then be likely. This removes a large factor that keeps these transitions from happening.

On top of this, I enjoy the fact that the implementation of this algorithm is nearly trivial on top of LB or GWG. One must simply add the path-length proposal and apply the LB/GWG proposal X times. Methods like this are likely to find quick impact since they can be easily incorporated into existing systems. This is compared to continuous relaxation-based discrete sampling methods which add many additional hyper-parameters, must be carefully tuned, and still tend not to perform as well as the methods discussed in this work.

I find the results presented here compelling. On a wide array of distributions, PAS (and PAFS) appear to outperform LB (and GWG) when evaluated in distinct ways. Further, there *appears* (see comments in Weaknesses section) to also be a notable improvement when using PAFS to train EBMs.

The work is not overloaded with theory (a good thing!). The authors provide what is necessary for understanding and correctness of the algorithm while leaving the work accessible to readers who are not experts in MCMC. Theorems 1 and 2 were easy to follow.

Overall, this method appears to be a solid step forward in MCMC for general discrete distributions, well supported with theory and empirical results.


Weaknesses:

A key elegance of LB/GWG is the lack of hyper-parameters to tune. MCMC hyper-parameters are notoriously difficult to tune and, at the same time, crucial for good performance. This work adds a path-length proposal distribution as a new hyper-parameter on top of those earlier works. Do the authors have any proposals for how to choose the path-length proposal? I can imagine we'd like to maximize something like the average hop distance. Do the authors have any thoughts on this? If X is set too large for example, then I would assume most proposals would be rejected. There has been much work on this in the context of HMC (to which this method has some spiritual similarities). Even if the authors do not have a concrete approach for choosing this distribution, some discussion of the desiderata of the optimal choice would be appreciated to help guide practitioners in the future.

Section 3.4 should contain a reference to Grathwohl et al. (2021) as this is clearly an extension of ideas presented in that work. As well, theorem 3 of this work is remarkably similar to theorem 1 of that work. This should be noted.

I find the deep EBM (and Ising training) results somewhat misleading. Which PAFS-X do you use? What is the runtime compared to GWG? How many steps are used per iteration in training? If you are using PAFS-X and N steps per iteration then you should compare with GWG using N*E[a(X)] steps since the runtime of PAFS-X is E[a(x)] times the runtime of GWG. How does this change the relative performance of the approaches? When training EBMs one can often get away with simpler MCMC approaches as the model tends to tune itself to the sampler used. This is why SGLD has had much more success in EBMs than more sophisticated samplers such as HMC. If the evaluation was more fair and the performance was still improved I'd be more likely to view this as a straight win over GWG for EBM training. If not, then there would still be some benefit to using GWG since it has no hyper-parameters to tune making it easier for practitioners less familiar with the nuances of tuning MCMC algorithms.

The authors should add an appendix section on experimental details for the EBM training. How many steps are used per iteration? Which PAFS-X is used? How are the log-likelihood scores evaluated? As it stands the results are not reproducible and I am not convinced of the improvements over GWG presented. Based on the other results in the paper, I believe the results are likely to hold but further evidence should be presented.

Add x-axis labels to figures 2 and 4.


**Summary Of The Paper:**

This paper proposes an MCMC sampling algorithm for discrete distributions. The algorithm can be thought of as an extension of Locally-Balanced proposals from Zanella where the sampler proposes local moves changing one dimension of the input at a time proportional to the likelihood difference of the model in the current state versus each proposed modified state. The key difference here is the authors propose to sample from this proposal distribution multiple times before the accept/reject step. This is somewhat analogous to the difference between Langevin Dynamics and Hamiltonian Monte-Carlo. Ideally, by applying the proposal distribution multiple times before the accept/reject step, the sampler is more likely to escape local modes than the original Locally Balanced algorithm.

The authors propose a fast variant of this algorithm which is an extension of the recently proposed Gibbs-With-Gradients sampler. This variant replaces the Locally Balanced proposal with the fast approximate version given by Gibbs-With-Gradients. This sampler approximately retains the benefits of the Locally Balanced proposal while being much faster to compute.

The authors demonstrate the performance of their sampler at sampling from a number of common discrete distributions and at training discrete Energy-Based Models. They find their sampler performs favorably when compared with Locally Balanced proposals, and Gibbs-With-Gradients.

--------------------- Post rebuttal period ------------------
I am very glad with the discussion that we've had and I thank the authors for responding to my critiques with new experiments and new results. I think they tell a more complete story of how the method relates to prior work. I believe we have cleared up many issues that other reviewers had with the work. I liked this work initially and I am glad now that other reviewers see it as I did. Having said that, I will keep my recommendation as it was initially. I would be very happy to see this work accepted.


**Summary Of The Review:**

This work presents a simple extension over prior state-of-the-art discrete MCMC methods which the authors demonstrate leads to improved performance on a wide variety of tasks. The method is simple, easy to implement, and appears correct. The authors provide the necessary theory for understanding the method and provide sufficient evidence for the performance of the method. There are some minor issues with the presentation of this evidence that should be addressed, particularly in the EBM training section. I would support acceptance of this work.

---

> ### Author Response · Authors · 2021-11-16
> **Response to Reviewer CSos**
>
> ### **Path Length**
> We add experiments in Ising, FHMM, and RBM to evaluate the performance of PAS and PAFS in terms of energy function evaluations under different path lengths. The results show that our PAFS, employing a soft no replacement sampling, obtains robust improvements across all path length. For this reason, when the computational bottleneck is evaluating the energy function, we can employ an aggressive policy to select a large path length for PAFS. To further improve the sampler, we believe one possible approach is connecting the Markov chain with a Markov jump process, such that we can substitute $\alpha(L)$ by $\alpha(L|x)$ depends on the holding time at current state $x$. Another possible approach is borrowing the idea from HMC and adaptive tuning the parameters. More discussion is given in section 6 in our revision.
>
> ### **About Section 3.4**
> We have added a comment in section 3.4 to point out the similarity and the difference between our PAFS and GWG. For theorem 3, We also add an experiment to demonstrate, though a simple modification, PAFS can perform significantly better than GWG when path length is large. The results can be found in figure 6 in our revision (or conveniently via link [Figure 6](https://i.ibb.co/8PcmDrX/fig6.jpg)).
>
>
> ### **Training Detail**
> We use the same setting as GWG, including the batch size, number of iterations, the PCD hyper parameters, etc.  In principle, both PAFS and GWG require two energy function evaluations per MCMC step, despite that PAFS explores much larger neighborhood space via path auxiliary.  As the energy function evaluation and gradient calculation usually dominates the computation, we expect these two should have similar run time. We report the average training time per batch of 100 chains in sec. B.5 in our revision.  For deep EBMs, PAFS with different path lengths run even slightly faster than GWG. Given that both of the methods implemented using pytorch, the minor speed up might be due to the implementation issue. So overall we think it is fair to say the two methods run equally fast in most cases. For this reason, we use the same number of steps when we train the deep EBMs. As is reported in GWG, we use the same number of steps (40) to train the deep EBMs. For our method, we also tune the expected path length in {3,5,7} and report the result with the best validation likelihood.
>
> For Ising-model learning: GWG runs 1.4x faster as the energy function computation now becomes cheaper. Following reviewer's advice, we allowed GWG to run 1.4x more sampling steps (denoted as GWG-eq in the figure) in each gradient update step and updated the figure 8.
>
> [Figure 8: Ising-model learning](https://i.ibb.co/D1sQkgc/fig8.jpg)
>
> ### **Figure**
> We have updated the figures with x-axis labels.

---

### Official Review · Reviewer_EvqS · 2021-10-28

**Correctness:** 4
**Technical Novelty And Significance:** 3
**Empirical Novelty And Significance:** 3
**Recommendation:** 8
**Confidence:** 5

**Main Review:**

**Contribution and Potential:**

The work tackles an important open problem for MCMC in discrete spaces, as it enables making large sampling steps while retaining the local benefits of pointwise informed proposals. This has important consequences, for instance when sampling from or training energy-based models. The result in Theorem 2 is to my knowledge a simple but non-trivial extension of the result in (Zanella 2020) and it’s significant in this context, as it provides an initial answer to the question “are balancing functions optimal on large neighborhoods?”. (Indeed, (Zanella 2020) focused on the case of L=1, while the authors consider the extended setting with L>=1.)

**Related Work**

Some recent work must be discussed. For instance, see (Dai 2020), (Jaini 2021) and (Sansone 2021).

**Questions:**

1. Theory. While certainly the result in Theorem 2 holds in the limit of infinite dimension. Is there a limit on the result of Theorem 2, or equivalently an upper bound on the value of L when the dimension of the space is finite? Indeed, as (Zanella 2020) points out, balancing proposals may not be optimal when considering global moves. Similarly, the constant in the Peskun ordering relation (in Theorem 2) should vanish when increasing L. Can the authors elaborate more on that?
2. Experiments. How is the maximum L determined and how can it be chosen in practice?
3. Experiments. How is the burn-in time chosen in all experiments?
4. Related Work. How does the proposed solution relate to existing recent work?

**Minor:**

Page 2 should $\{0,1\}^p$ be replaced by $\{0,1\}^n$, in order to be consistent with what follows?

Page 7 Figure 10 -> Figure 3

Spacing between text and citations

**References**

(Zanella 2020) Informed Proposals for Local MCMC in Discrete Spaces

(Grathwohl 2021) Oops I Took a Gradient: Scalable Sampling for Discrete Distributions

(Dai 2020) Learning Discrete Energy-based Models via Auxiliary-variable Local Exploration

(Jaini 2021) Sampling in Combinatorial Spaces using SurVAE Flow Augmented MCMC

(Sansone 2021) LSB: Local Self-Balancing MCMC in Discrete Spaces

**Summary Of The Paper:**

The paper provides a generalisation of the work of pointwise informed proposals (Zanella 2020), where the acceptance criterion is applied after L sampling steps, instead of each single one.
This comes with the advantage that the overall acceptance criterion  is guaranteed to be larger than the corresponding one in (Zanella 2020). The authors also extend the results in (Zanella 2020), by proving under mild conditions that weighting functions based on balancing ones are asymptotically optimal as for sampling efficiency according to Peskun ordering. Lastly, they provide an algorithmic extension reducing the number of target likelihood evaluations by applying the approximation strategy proposed in (Grathwohl 2021). The analysis is corroborated by experiments on several energy based models both for inference and learning tasks, thus showing the superiority of the proposed algorithm and its extension over existing MCMC strategies.


**Summary Of The Review:**

**Initial recommendation**

The paper is well written and technically sound. I went through the proofs and confirm their validity.
The paper represents an added value to the existing body of knowledge on MCMC, especially in the discrete domain. However, I have some questions for the authors, which might affect the final result. I initially recommend for a weak accept.

**Final recommendation**

The authors have addressed all concerns during the rebuttal. Therefore, I increase my score.

---

> ### Author Response · Authors · 2021-11-16
> **Response to Reviewer EvqS**
>
> ### **Limit Results for Theorem 2**
> Yes, we can have the limit results for theorem 2. Using Peskun ordering, we have $P_{\bar{g}}(x, y) \ge C^{(n)} P_g(x,y)$, where asymptotic ratio $1 \ge C^{(n)} \ge 1 - \mathcal{O}(\frac{U d_n}{n})$. More detailed discussion can be found in the newly added Appendix A.4.
>
> ### **Choice of $L$**
> Selecting the path length is a challenging problem as tuning parameters is notoriously difficult in MCMC.
> We add experiments in Ising, FHMM, and RBM to evaluate the performance of PAS and PAFS in terms of energy function evaluations under different path lengths. The results show that our PAFS, employing a soft no-replacement sampling, obtains robust improvements across all path length. For this reason, when the computational bottleneck is evaluating the energy function, we can employ an aggressive policy to select a large path length for PAFS. To further improve the sampler, we believe one possible approach is connecting the Markov chain with a Markov jump process, such that we can substitute $\alpha(L)$ by $\alpha(L|x)$ depends on the holding time at current state $x$. Another possible approach is borrowing the idea from HMC and adaptively tuning the parameters. More discussion is given in section 6 in our revision.
>
> ### **Burn-in Time**
> In this work, we use the first half of the chain as burn-in time to compute ESS. Though some chains have fast mixing, some other chains need much longer time for mixing. We choose a long burn-in time to make sure all chains reach their stationary distributions.
>
> ### **Related Work**
> Thanks for pointing out the missed related work. [Dai 2020] introduces the path as a latent variable in the variational distribution for initializing PCD, but still relies on slow Gibbs sampling for improvement. [Jaini 2021] uses continuous relaxation. It first learns continuous embedding of discrete space via SurVAE flow, which allows to simulate MCMC chains in the continuous space and then map the chain back into discrete space. The limitation of continuous embedding is that it can destroy the natural topological properties of the discrete space under consideration.  [Sansone 2021] has a different focus to our work. It parameterizes the locally balanced function and adapts the parameters by maximizing the mutual information to select a good weight function among the family of locally balanced functions. We have added them to the related work in our revision.

---

> > ### Comment · Reviewer_EvqS · 2021-11-21
> > **Limits of Theorem 2/Burn-in time**
> >
> > Thanks for the answer and the added parts in the document, such as appendices A.4, A.5!
> >
> > **Theorem 2**
> >
> > Eq. (37)  in the appendix depends on the connectivity of the independence graph $\frac{d_n}{n}$, on the balancing function $g$ and the likelihood ratio $\frac{\pi(y)}{\pi(x)}$, which influences the performance in terms of efficiency according to Peskun ordering. In practice (i.e. finite dimension), we may still have cases in which the sampler is suboptimal. What could happen when dealing with densely connected independence graphs, or distributions characterized by highly-varying energy functions, or distributions with modes separated by large low-mass regions?  In such cases, you could experience a drop in performance when choosing a large path length (as L is exponentiating the $c_g$ constant in Eq. (52)).
> >
> > **Burn-in Time**
> >
> > What does it mean "In this work, we use the first half of the chain as burn-in time to compute ESS". Do you compute ESS during burn-in?

---

> > > ### Author Response · Authors · 2021-11-21
> > > **Responses**
> > >
> > > ### **Theorem 2**
> > > Yes, our asymptotic results depend on the connectivity of the independence graph $\frac{d_n}{n}$. In the finite dimension regions, we need more sophisticated characterizations of the dependence of the model. For example, a pairwise influence matrix
> > > $\Psi_\pi(i, j) := \pi(v_j=1|v_i=1) - \pi(v_j=1|v_i=0)$
> > > can describe the dependence in a binary model. It does not require hard independence between variables in the model but gives a quantitative characterization of how variables depend on other variables.  People make different assumptions on the influence matrix, for example, negative dependence [FM92] or spectral independence [ALO20], to derive different bounds. Using these more sophisticated assumptions, we can improve the ratio in Equations 41, 42, 43 in lemma 2, thereby the ratio in Equation 52. However, such analysis is beyond the scope of this work, hence we remain to use the simpler assumption, degree in independence graph, to introduce our path auxiliary algorithm. Despite the bounds not being sharp, our algorithms perform well across many models, with path lengths up to 20 (though in RBM using length longer than 10 won’t help further), in our experiments. The results imply there is a large room to improve the bounds under more sophisticated assumptions and we will explore it in our future work. We will add this discussion to our appendix accordingly if you think this could help address your concerns.
> > >
> > >
> > > ### **Burn-In**
> > > Sorry for the confusion here. What we meant is we use the first half of the chain as burn-in, and use the last half of the chain to compute ESS. For example, in RBM, we use the first 50,000 MCMC steps as burn-in and use the last 50,000 steps to compute ESS. From the trace plots (see e.g., first 10,000 MCMC steps in RBM
> > > [[Figure 20: GWG]](https://i.ibb.co/2Kv1yXh/results-gwg.jpg) and [[Figure 21: PAFS]](https://i.ibb.co/hVYnS8C/results-pa.jpg)) we can see these many steps seem to be long enough for the mixing. You can also find more burn-in curves for PAS and PAFS in different models in appendix B.
> > >
> > > >[FM92] Balanced Matroids\
> > > [ALO20] Spectral Independence in High-Dimensional Expanders and Applications to the Hardcore Model

---

> > > > ### Comment · Reviewer_EvqS · 2021-11-24
> > > > **Thanks**
> > > >
> > > > Thanks for the clarifications.
> > > >
> > > > It's not necessary to add the discussion about sharpening the bounds in theorem 2. My concern was more conceptual as you are considering the use of balancing functions (known to be optimal in the context of local sampling) in a "more global" one. Therefore , I was wondering if there is a limit on the increase of the path length to ensure the optimality of the balancing function. However, the theorem 2 and your modified appendix already provide insights about the answer to that. Indeed, you can't increase the path length indefinitely and I would expect that, even if you sharpen the bounds, they would still depend exponentially on the path length. In any case, this was a mere consideration.
> > > >
> > > > Overall, I like your work and I'm happy to champion it by updating my score.

---

### Official Review · Reviewer_Eoj5 · 2021-10-30

**Correctness:** 3
**Technical Novelty And Significance:** 3
**Empirical Novelty And Significance:** 3
**Recommendation:** 8
**Confidence:** 3

**Main Review:**

Locally informed proposals and the use of locally balanced functions were proposed by Zanella (2020). The paper uses these ideas in a new way to efficiently explore large neighborhoods via auxiliary paths.

Strengths:
1. The method proposed is more efficient (in terms of likelihood evaluations) than naively expanding the neighborhood using a locally-informed proposal.
2. Using a balanced function the resulting acceptance rate takes a very simple form and can be computed efficiently.
3. Experiments include many models and several baselines (though I have some comments on the specifics).

Weaknesses:
1. I find section 3.3 a bit confusing. A few concrete comments: (a) Where does the definition of an "ideal" function come from? Theorem 2 states that, for every ideal function f, there is a locally balanced function g asymptotically better than f. Why does this mean asymptotic optimality for locally balanced functions? Am I missing something simple? (b) Is the conditional independence graph (thm 2) defined anywhere? Is it the DAG that defines the factorization of the distribution as in a directed graphical model?

2. The result from theorem 3 (about the use of the Taylor expansion) and its analysis seem problematic. Specifically the part that claims that the use of the approximation does not lead to a large detrimental effect. The theorem lower bounds the efficiency lost by using the approximation. But this lower bound decreases *exponentially* with K and with $U^2$, where U is the length of the path. If U=10 this bound says that using the Taylor approximation leads to a method whose efficiency is at least $T\times e^{-50}$, where $T$ is the efficiency of the method without the approximation. Claiming that the approximation does not lead to "losing too much proposal quality" does not seem justified. It could be the case that this bound is very lose, and thus not informative. Or it could be that there are models for which the inequality becomes an equality, and the use of the Taylor approximation becomes extremely problematic (would it be possible to find a simple model that satisfies this?). Do you have any comments on this? I am not claiming the approximation is necessarily bad, I'm just asking about this specific theoretical result and its relevance.

Finally, I have some comments on the empirical evaluation. First, I'd like to mention that I find the wide variety of models, tasks and baselines very nice. However, I think the section can be improved by showing more organized experiments addressing each component of the proposed method separately. Concretely:

E1. Section 3.2 shows that the method with a path of length L is more efficient than running LB-1 for L steps. It would be nice to have experiments that show the effect of this in practice. This could be done by comparing PAS-L and LB-1 on several models, in terms of likelihood evaluations, not MCMC steps (PAS-L uses L times as many likelihood evaluations per step). Figs 1 and 2 compare these methods in terms of MCMC steps on two simple models. I think the results in terms of likelihood evaluations would be similar. However, these models, while simple and illustrative, may be too tailored to make LB-1 fail. It would be great to see results for the Ising model, RBM, FHMM.

E2. Effect of introducing the Taylor approximation. This could be seen by compare PAS-M vs PAFS-M for a bunch of models, for varying M, in terms of MCMC steps (not likelihood evaluations). While not a fair comparison, this reflects the effect the approximation has, and how much we are losing by using it. (In this setup, we can only expect that using the approximation will hurt performance. This would show how much.) For a practical comparison, results in terms of likelihood evaluations should be shown too (and are currently shown). Also, for the real models, PAS and PAFS are compared only for the Ising model, which the authors state is close to linear (and thus possibly favors the Taylor expansion approximation).

E3. Comparison of the final method with the Taylor approximation against other baselines (GWG-M, LB, Gibbs, etc) in terms of likelihood evaluations.


Additional comments:
- Should add labels (in x and y axis) in all plots (e.g. x-axis in Fig 4, is it likelihood evaluation? MCMC steps?).
- Plots in Fig. 3 would be nice to have in same scale for the y-axis.
- Is there a typo in equation 10? Should it be p instead of n in the summation?

**Summary Of The Paper:**

MCMC in discrete spaces using locally informed proposals usually use small neighborhoods (e.g. changing only one component), for efficiency reasons. The paper proposes a method to efficiently explore bigger neighborhoods using locally informed proposals (with a locally balanced function) to sample "auxiliary paths". Additionally, following previous work, they propose an efficient version using a linear Taylor expansion, thus saving a significant number of likelihood evaluations.

**Summary Of The Review:**

On the positive side, I think the paper provides a nice and natural extension of MCMC methods in discrete spaces based on locally informed proposals.

On the negative side, I think that some of the theory could be presented more clearly, some of the claims revisited, and the empirical evaluation could be modified a bit to better show the benefits/drawbacks of each of the method's components.

-----

I updated my score after the authors' response.

---

> ### Author Response · Authors · 2021-11-16
> **Response to Reviewer Eoj5 (Part I)**
>
> ### **Ideal Function**
> We define the ideal function as a stepping stone to show the asymptotic optimality of locally balanced function. We name it *ideal* as the its properties are ideal assumption for a weight function in PIP. Conditions 1) $f: \mathbb{R}^+ \mapsto \mathbb{R}^+$  and condition 2) $f(1)= 1$ are natural requirements for a weight function. Condition 3) where $f(t)$ and $f(t)f(\frac{1}{t})t$ are monotonically increasing, though, is a technical requirement, it is reasonable for weight functions.  The asymptotic optimal in theorem 2 here refers to the asymptotic optimal within the ideal function $\mathcal{F}$. The ideal function class contains most of commonly used weight function, such as $f(t) = \frac{2t}{1+t}, f(t) = \max\{1, t\}, f(t) = \min\{1, t\}$, and $f(t) = t^\alpha, \alpha \ge 0$. Also, it is convex in the log space, and is a superset for normalized monotonically increasing locally balanced functions. For these reasons, we say theorem 2 strongly suggests that locally balanced function is a good choice for path auxiliary sampler. To further clarify this, we have updated Appendix A.3 on more details and discussion for ideal function.
>
> ### **Conditional Independence Graph**
> The (undirected) conditional independence graph corresponds to Markov random fields [1]:
>
> > Definition 1. *The conditional independence graph of $X$ is the undirected graph $G = (K, E)$, where $K = \{1, ..., K\}$ and $(i, j) \notin E$ if and only if $X_i \perp X_j | X_{K \backslash \{i, j\}}$.*
>
> We have added this definition in Appendix A.3
>
> ### **Approximation via Taylor's expansion**
> You are correct, the bound in theorem 3 is loose as we analyze it under the worst case.
> Though the theorem is not sharp, it provides a framework to prove approximation bound for path auxiliary sampler via single step approximation. More sophisticated bounds can be obtained by using conditional independence.
> For example, in the Ising model, an auxiliary path has a high probability to avoid manipulating two adjacent nodes, and in this case, the linearization is accurate and we lose nothing in approximation. Such analysis relies on a more accurate assumption of conditional independence and we will leave it to our future work. More discussion can be found in appendix A.7.

---

> ### Author Response · Authors · 2021-11-16
> **Response to Reviewer Eoj5 (Part II)**
>
> ### **Experiments**
> Thanks for the great suggestions on the experiments section. Following your advice, we have added the experiments you suggested in our revision.
>
> **E1.** For comparing PAS and LB, we updated figure 1 and figure 2 in terms of energy evaluations, and as you foresaw, the results still strongly support the effectiveness of PAS. We paste the figures here for convenience:
>
> [[Figure 1: Estimation Error of Distribution Mean on Parity Model]](https://i.ibb.co/GHfcvPG/fig1.jpg)
> [[Figure 2: Energy Trace on Weighted Permutation Model]](https://i.ibb.co/S3F1PxT/fig2.jpg)
>
>
> We also evaluate PAS-1, PAS-2, PAS-3, PAS-5, PAS-10 in Ising and FHMM (PAS-1 is equivalent with LB). In the Ising model, the additional results are reported in the updated Appendix B.1. For convenience we've included the results on Ising models with difference sizes here:
>
> [[Figure 9: Ising-50]](https://i.ibb.co/1qPpvfT/ising-50.jpg) [[Figure 10: Ising-100]](https://i.ibb.co/rpx6jLk/ising-100.jpg) [[Figure 11: Ising-150]](https://i.ibb.co/jwX2HKw/ising-150.jpg) [[Figure 3: Ising-200]](https://i.ibb.co/47XMc0s/ising-200.jpg)
>
> The ESS in terms of MCMC steps monotonically increases when we use larger path lengths. The ESS in terms of energy function evaluations obtains slight increase and reaches its peak at PAS-5.
>
> In FHMM model, the additional results are reported in the updated Appendix B.2. We also include the results on different models here:
>
> [[Figure 15: FHMM-500]](https://i.ibb.co/0BGPzrG/fhmm-500.jpg) [[Figure 16: FHMM-1000]](https://i.ibb.co/YLKLHVS/fhmm-1000.jpg) [[Figure 17: FHMM-1500]](https://i.ibb.co/K6nqwHt/fhmm-1500.jpg) [[Figure 18: FHMM-2000]](https://i.ibb.co/JqvZXth/fhmm-2000.jpg) and [[Figure 19: ESS comparison]](https://i.ibb.co/d0SMCVz/fhmm-compare.jpg)
>
> While increasing the path length is helpful in mixing, however, it can not improve ESS in terms of energy function evaluations. The results show whether PAS is helpful depends on the property of the target distribution.
>
>
> **E2.** To show the effect of introducing the Taylor approximation, we added comparison between PAS-1, PAS-2, PAS-3, PAS-5, PAS-10 and PAFS-1, PAFS-2, PAFS-3, PAFS-5, PAFS-10 correspondingly in Ising and FHMM. In the Ising model, the results are reported in Appendix B.1 (or the above figures: [[Figure 9: Ising-50]](https://i.ibb.co/1qPpvfT/ising-50.jpg) [[Figure 10: Ising-100]](https://i.ibb.co/rpx6jLk/ising-100.jpg) [[Figure 11: Ising-150]](https://i.ibb.co/jwX2HKw/ising-150.jpg) [[Figure 3: Ising-200]](https://i.ibb.co/47XMc0s/ising-200.jpg))
> For the ESS in terms of MCMC steps, PAFS has very similar performance as PAS, especially when the problem size is large. For ESS in terms of energy function evaluations, PAFS significantly outperforms PAS. In FHMM, the results are reported in Appendix B.2 (or the above figures: [[Figure 15: FHMM-500]](https://i.ibb.co/0BGPzrG/fhmm-500.jpg) [[Figure 16: FHMM-1000]](https://i.ibb.co/YLKLHVS/fhmm-1000.jpg) [[Figure 17: FHMM-1500]](https://i.ibb.co/K6nqwHt/fhmm-1500.jpg) [[Figure 18: FHMM-2000]](https://i.ibb.co/JqvZXth/fhmm-2000.jpg) and [[Figure 19: ESS comparison]](https://i.ibb.co/d0SMCVz/fhmm-compare.jpg)). In terms of MCMC steps, PAFS has poorer approximation of PAS compared to the results in Ising as FHMM's energy function is farther from linear.
>
> **E3.** For comparing PAFS with other samplers, we add a comparison with GWG in sampling from RBM trained on MNIST. We report their accept rate, hop distance, and ESS under different path lengths in [Figure 6](https://i.ibb.co/8PcmDrX/fig6.jpg) in our revision. The results show that our PAFS, employing a soft no replacement sampling strategy, is robust in path length, and significantly outperforms GWG.
>
> We tried to reflect the above logic in the updated text, while keeping the current ordering of the studies. Most of the results for the experiments discussed above are added in appendix, and we hope the updated experiments would resolve your concerns.
>
> > [1] (Whittaker 1990) Graphical Models in Applied Multivariate Statistics, p. 60

---

> > ### Comment · Reviewer_Eoj5 · 2021-11-18
> > **Thanks!**
> >
> > Thanks for the detailed response and clarifications. I think the figures attached are extremely informative. I'm happy to increase my score, I really like this work.
> >
> > A final suggestion I have would be to update the wording after theorem 3. It is fine to leave the detailed discussion to the Appendix, but as it is now the "without losing too much proposal quality" is not justified. I'm fine with the analysis staying as it is (the empirical results are great), but the main paper should include a discussion about the tightness. Furthermore, the discussion in the Appendix states that better bounds may be possible for some independence structure, but not for others. This appears to indicate that there are cases (maybe adversarial) for which the bound cannot be improved much. I think that adding a couple sentences about this in the main paper would be good. One may even add that despite this loose/unfavorable result, the method works very well in practice (with a forward reference to the experiments).

---

> > > ### Author Response · Authors · 2021-11-18
> > > **Thanks!**
> > >
> > > Dear reviewer, thank you for your recognition of our work. Following your suggestion, we have added sentences discussing the tightness of Theorem 3 in section 3.4 in the latest revision. Thanks for your help in improving our work!

---

### Official Review · Reviewer_y5iJ · 2021-11-03

**Correctness:** 4
**Technical Novelty And Significance:** 3
**Empirical Novelty And Significance:** 3
**Recommendation:** 8
**Confidence:** 5

**Main Review:**


## Weakness

1 - The primary weakness in the paper is that the proof of Theorem 1 is incorrect. Equation (17) can be split into two parts:

A - $\pi(x)\sum_{L} \alpha(L) \sum_{(\sigma,L) \in \Sigma(\mathcal{X}, \mathcal{N}) \colon \sigma_0 = x, \sigma_L = y} \Big[ \prod_{l=1}^L Q_0(\sigma_{l-1},\sigma_l)\Big]$

B - $\min{} \Big[1, \frac{\pi(y)\prod_{l=1}^L Q_0(\sigma_l, \sigma_{l-1})}{\pi(x)\prod_{l=1}^L Q_0(\sigma_{l-1}, \sigma_{l})}\Big]$

While part A marked above is acurate, part B should technically be  $\min{} \Big[      1, \frac{\pi(y)\prod_{l=1}^{L'} Q_0(\delta_l, \delta_{l-1})}{\pi(x)\prod_{l=1}^{L'} Q_0(\delta_{l-1}, \delta_{l})} \Big]$

where $L'$ and $\delta$ are the length samples and path samples drawn in the algorithm mentioned above Theorem 1.
As such Equation 17 doesn't imply Equation 18.

Despite this theoretical inconsistency, it is not surprising that the experimental section shows that the proposed method works better than [GWG].
This is because the method is a Pseudo Marginal MH procedure (where the acceptance probability is a Monte Carlo sample of the true acceptance probability like [Section 4.2, RB]).
If so presented, I would be inclined to accept the paper but in the current form, the paper is mathematically incorrect.

2 - The second listed contribution (linearization) is a rehash of [GWG] and is not novel.

3 - There are multiple grammatical errors in the paper examples include:
  - The first phrase of the abstract should've been `EBMs offer`
  - Second sentence, second para, page 3 should be `Instead of directly sampling from a large neghborhood path auxiliary sampler sequentially samples new states from a ...`.

4- Minor question: In the thrid paragraph, do you mean a simple random walk? If so what is the graph over which the RW takes place? Did you mean Gibbs sampling?

## Stregths

1 - The experimental section is very strong and the explanation in terms of the toy example is very intuitive.

[GWG] - Will Grathwohl, Kevin Swersky, Milad Hashemi, David Duvenaud, and Chris J Maddison. Oops i took a gradient: Scalable sampling for discrete distributions. arXiv preprint arXiv:2102.04509, 2021.
[RB] - On Markov chain Monte Carlo methods for tall data
R Bardenet, A Doucet, C Holmes - The Journal of Machine Learning Research, 2017 https://www.jmlr.org/papers/volume18/15-205/15-205.pdf


**Summary Of The Paper:**

- Consider the task of MH-MCMC on a EBM defined over discrete state space.
- [GZ] proposed a Kernel which evaluates the energy function over a Hamming ball surrounding the current state and [GWG] improved the efficiency of the latter by using a Taylor series expansion of the energy function.
- The authors of the current paper improve the acceptance ratio (Eq 2) and mixing (Thm3) by sampling a multiple transitions (a path) instead of a single neighborhood sample.

[GZ] - Giacomo Zanella. Informed proposals for local mcmc in discrete spaces. Journal of the American Statistical Association, 115(530):852–865, 2020.

[GWG] - Will Grathwohl, Kevin Swersky, Milad Hashemi, David Duvenaud, and Chris J Maddison. Oops i took a gradient: Scalable sampling for discrete distributions. arXiv preprint arXiv:2102.04509, 2021.


**Summary Of The Review:**


The proof of the main theorem is incorrect and hence I am rejecting the paper.

--------------
I am updating my scores based on author discussion. Please refer to comments for details.

---

> ### Author Response · Authors · 2021-11-16
> **Response to Reviewer y5iJ**
>
> ### **Proof for Theorem 1**
> We thank the reviewer for checking the proof. However, we believe there may be some confusion on the notation used causing the potential misunderstanding. We clarify that $\pi(x)K(x, y)$ should be expanded in the following way:
>
> $\pi(x) \sum_{L} \alpha(L)  \underbrace{\sum_{\substack{(\sigma, L) \in \Sigma(\mathcal{X}, \mathcal{N}):\\ \sigma_{0}=x, \sigma_{L}=y}} \underbrace{\Bigg[\overbrace{\Big(\prod_{l=1}^L Q_0(\sigma_{l-1}, \sigma_{l})\Big)}^\text{Probability for path $\sigma$ condition on length $L$}\overbrace{\min \left\\{ 1, \frac{\pi(y) \prod_{l=1}^L Q_0(\sigma_{l}, \sigma_{l-1})}{\pi(x) \prod_{l=1}^L Q_0(\sigma_{l-1}, \sigma_{l})} \right\\}}^\text{Accept Rate}\Bigg]}_\text{Probability for transition from $x$ to $y$ through path $\sigma$ condition on length $L$}}_\text{Probability for transition from $x$ to $y$ condition on length $L$}$
>
> In our method, we introduced the auxiliary path to augment the proposal distribution (rather than the target distribution $\pi$). So the above equation calculates the transition kernel via integration over all possible auxiliary paths.We hope that the above equation, with the relevant terminology highlighted, is easier to follow. We've also updated Appendix A.1 for the full proof of the detailed balance property. We would like to kindly ask the reviewer to check the updated proof and see if you agree on the mathematical correctness of the theorem.
>
> ### **Contribution for Linearization**
> We clarify that the linearization used in PAFS is inspired by GWG and is very similar. However, we want to highlight the subtle but important difference here: GWG draws all indices from the linearization at $\sigma_0$. As a result, GWG may repeatedly select the same index thereby reducing the efficiency, especially after mixing. Our PAFS sampler uses the linearization at *each* state $\sigma_l$ and effectively reduces the backtrack. Though a simple modification, PAFS shows substantial improvements compared to GWG. To demonstrate this, we added a comparison between PAFS and GWG on sampling from RBM. The results are reported in figure 6 in our revision. For convenience,  we include the figure links here as well.
>
> [Figure 6: PAFS-M v.s. GWG-M for different \# edits M](https://i.ibb.co/8PcmDrX/fig6.jpg)
>
> GWG suffers a lot when we use large path lengths, while our PAFS obtains robust improvements under different path lengths. We want to emphasize that simply using sampling without replacement for GWG would not help as it results in an extreme long mixing as shown in figure 22 in our revision, or the figure below:
>
> [Figure 22: sampling without replacement in GWG-M mixes slower](https://i.ibb.co/8dRPJJt/fig22.png)
>
> ### **Random Walk**
> The terminology of *random walk* mentioned in paragraph 3 means an uninformed proposal or says a uniform proposal from the neighborhood. It is not Gibbs sampling. In the language of pointwise informed proposal (PIP), random walk means using a uniform weight function $g(x) \equiv 1$.
>
> ### **Grammatical errors**
> Thank you for pointing out the errors in writing. We have fixed them in our revision according to your suggestions.

---

> > ### Comment · Reviewer_y5iJ · 2021-11-16
> > **Regarding the proof for Theorem 1**
> >
> > Dear authors,
> > I apologize for not spending more time clarifying my concern regarding Theorem 1.
> > I would like to elaborate as follows:
> >
> > $K(x,y)$  from Equation 17 (which is the transition probability of the Markov chain) can be decomposed as $Q(x,y) a(x,y)$, where $Q$ is the proposal distribution and $a$ is the acceptance probability.
> >
> > Both of us agree that the proposal distribution is given by $Q(x,y) = \sum_{L} \alpha(L) \sum_{(\sigma,L) \in \Sigma(\mathcal{X}, \mathcal{N}) \colon \sigma_0 = x, \sigma_L = y} \Big[ \prod_{l=1}^L Q_0(\sigma_{l-1},\sigma_l)\Big]$ which accounts for all possible ways of sampling $y$ given the current state $x$.
> >
> > According to the procedure described (Section 3.1), however, the acceptance probability solely depends on the path that was sampled in the MCMC procedure, which is:
> >  - A path length $L'$ is sampled and then a path $\delta$ is sampled to yield the proposal $y$.
> >  - $a(x,y)$ is then calculated as $\min{} \Big[      1, \frac{\pi(y)\prod_{l=1}^{L'} Q_0(\delta_l, \delta_{l-1})}{\pi(x)\prod_{l=1}^{L'} Q_0(\delta_{l-1}, \delta_{l})} \Big]$.
> >
> > Note that this doesn't account for the various paths that can be followed to sample $y$. As such the products of path probabilities don't cancel out between equation 17 and 18.
> > The correct acceptance probability for this procedure would account for all path lengths and paths that go from x to y and as such would be infeasible.

---

> > > ### Comment · Reviewer_y5iJ · 2021-11-16
> > > **Correct acceptance probability**
> > >
> > > The correct acceptance probability should have been:
> > >
> > > $a(x,y) = \frac{\pi(y)\sum_{L} \alpha(L) \sum_{(\sigma,L) \in \Sigma(\mathcal{X}, \mathcal{N}) \colon \sigma_0 = y, \sigma_L = x} \Big[ \prod_{l=1}^L Q_0(\sigma_{l-1},\sigma_l)\Big]}{\pi(x)\sum_{L} \alpha(L) \sum_{(\sigma,L) \in \Sigma(\mathcal{X}, \mathcal{N}) \colon \sigma_0 = x, \sigma_L = y} \Big[ \prod_{l=1}^L Q_0(\sigma_{l-1},\sigma_l)\Big]}$

---

> > > ### Comment · Reviewer_y5iJ · 2021-11-16
> > > **Toy Example**
> > >
> > > Consider the following toy example:
> > >
> > > - Let there exist 2 paths between $x$ and $y$, via $u$ and $v$.
> > > - Let $\alpha$ be a dirac delta on $2$.
> > > - Let $x \to u \to y$ have unnormalized path weights $4$ and $5$. Similarly let $y \to u \to x$ have weights $2$ and $7$
> > > - Let $x \to v \to y$ have unnormalized path weights $3$ and $7$. Similarly let $y \to v \to x$ have weights $4$ and $2$.
> > > - Let $\pi(x) == \pi(y)$.
> > >
> > > The proposal distribution is $Q(x,y) \propto 20 + 21$ and the acceptance should have been $min(1, \frac{14+8}{20+21})$, however the procedure mentioned in the paper will either compute it as $min(1, \frac{14}{20})$ or $min(1, \frac{8}{21})$ depending on the path taken.

---

> > > ### Author Response · Authors · 2021-11-16
> > > **Response for the Proof of  Theorem 1**
> > >
> > > Dear reviewer, thank you so much for your detailed response. We think what you state is a marginal sampler. However, in our algorithm, we use an auxiliary sampler. They are different, but both of them can satisfy the detailed balance condition, though a marginal sampler could be intractable in many scenarios. Our following arguments are based on the results from [Liang et.al, Page 86](http://213.230.96.51:8090/files/ebooks/Matematika/Liang%20F.,%20et%20al.%20Advanced%20Markov%20chain%20Monte%20Carlo%20methods%20(Wiley,%202010)(ISBN%200470748265)(O)(379s)%20MVspa%20.pdf). Specifically,
> > >
> > > We augment the proposal distribution $T(y|x)$ by an auxiliary variable $u$, such that $T(y|x) = \int T_1(u|x)T_2(y|x,u)du$.
> > >
> > > - For a marginal sampler, the accept rate is what you mentioned: $A_\text{mar}(x, y) = \min\\{1, \frac{\pi(y)\int T_1(u|y) T_2(x|y, u)du}{\pi(x)\int T_1(u|x)T_2(y|x, u)du}\\} = \min\\{1, \frac{\pi(y)T(x|y)}{\pi(x)T(y|x)}\\}$\
> > > And the transition kernel marginal sampler is:\
> > > $K_\text{mar} (x, y) = T(y|x) A_\text{mar}(x, y)$
> > > but such a marginal sampler is intractable, as it requires to integrate the auxiliary path in both proposal and the accept rate calculation.
> > >
> > > - For a (proposal augmented) auxiliary sampler, given x, we sample auxiliary $u \sim T_1(u|x)$ and  new state $y \sim T_2(y|x, u)$. The accept rate is: \
> > > $A_\text{aux}(x, y, u) = \min \\{1, \frac{\pi(y) T_1(u|y)T_2(x|y, u)}{\pi(x)T_1(u|x)T_2(y|x, u)}\\}$\
> > > In this case, we need to integrate the auxiliary variable $u$ to obtain the transition kernel from $x$ to $y$. $K_\text{aux}(x, y) = \int_u T_1(u|x) T_2(y|x, u) A_\text{aux}(x, y, u) du$ (Equation (**4.1**) in Liang et.al)\
> > > Then following Equation (**4.2**) in Liang et.al we satisfied the detailed balance condition. By employing the auxiliary sampler, we avoid integrating the auxiliary path.
> > >
> > > Even using the same $T_1$ and $T_2$, marginal sampler and auxiliary sampler can result in different transition kernels. Our algorithm is not an approximation of the marginal sampler, but an auxiliary sampler. Let us know if this makes sense to you and we are happy to have deeper discussions on it.
> > >
> > > **Reference**
> > > > Liang et.al, Page 86, [Advanced Markov Chain Monte Carlo Methods](http://213.230.96.51:8090/files/ebooks/Matematika/Liang%20F.,%20et%20al.%20Advanced%20Markov%20chain%20Monte%20Carlo%20methods%20(Wiley,%202010)(ISBN%200470748265)(O)(379s)%20MVspa%20.pdf).

---

> > > > ### Comment · Reviewer_y5iJ · 2021-11-17
> > > > **Clarifying question.**
> > > >
> > > > I completely agree that $K_{\text{aux}}$ as you described above keeps the steady state invariant, however I am not able to understand the following:
> > > >
> > > > In the above set of equations I am referring to $K_{\text{aux}}$ in particular, and we can assume that $\alpha(L)$ is dirac-delta.
> > > >
> > > > Is $u$ the sampled path? If that is the case, the acceptance probability in the paper accounts for $\pi(x)$, $\pi(y)$ (obviously), $T_2(x|y,u)$ and $T_2(y|x,u)$ (via the product of the $Q_0$ over the path.)
> > > >
> > > > Do you assume that $T_1(u|y) = T_1(u|x)$? I.e. that the probability of sampling the path from $x$ to $y$ is the same as sampling the same path in reverse? The toy example I provided serves as a counter example to this assumption.
> > > >
> > > > If not, could you please help me understand the association between $K_{\text{aux}}$ and the acceptance probability used in your procedure i.e. what parts of the acceptance probability you use correspond to $T_1$, $T_2$, etc..
> > > >
> > > > Thank you for taking the time to go into detail, furthermore I apologize for not correctly identifying this as an auxiliary variable MCMC. However my confusion stemmed from the fact that this was not mentioned in the proof and because of the question I have asked in this comment.

---

> > > > > ### Author Response · Authors · 2021-11-17
> > > > > **Clarifying the transition kernel**
> > > > >
> > > > > Dear reviewer, thanks for your patience in discussion with us. In our sampler, as you mentioned, the auxiliary variable $u$ is the sampled auxiliary path $\sigma$. If we consider $\alpha(L)$ is dirac-delta, we have
> > > > > - $T_1(u|x) = T_1(\sigma | x) = \prod_{l=1}^L Q_0(\sigma_{l-1}, \sigma_l)$
> > > > > is the probability to sample the auxiliary path $\sigma$ condition on $x$.
> > > > > - $T_2(y|x, u) = \mathbb{I}\\{x, y \text{ are the two ends of path } u (\text{or say} \sigma) \\}$
> > > > > is deterministic, or say dirac-delta.
> > > > >
> > > > > Going back to our theorem, the equation, with a little abuse of notation, can be explained as:
> > > > >
> > > > > $\pi(x) \sum_{L} \alpha(L)  \underbrace{\sum_{\substack{(\sigma, L) \in \Sigma(\mathcal{X}, \mathcal{N}):\\ \sigma_{0}=x}} \underbrace{\Bigg[\overbrace{\Big(\prod_{l=1}^L Q_0(\sigma_{l-1}, \sigma_{l})\Big)}^\text{$T_1(\sigma|x, L)$} \overbrace{ \mathbb{I}\\{\sigma_L = y\\}}^\text{$T_2(y|x, \sigma, L)$} \overbrace{\min \left\\{ 1, \frac{\pi(y) \prod_{l=1}^L Q_0(\sigma_{l}, \sigma_{l-1}) \mathbb{I}\\{\sigma_0 = x\\} }{\pi(x) \prod_{l=1}^L Q_0(\sigma_{l-1}, \sigma_{l}) \mathbb{I}\\{\sigma_L = y\\} } \right\\}}^\text{$A(x, y, \sigma|L)$}\Bigg]}_\text{$T_1(\sigma|x, L) T_2(y|x, \sigma, L) A(x, y, \sigma|L)d\sigma$}}_\text{$K(x,y|L) = \int_\sigma T_1(\sigma|x, L) T_2(y|x, \sigma, L) A(x, y, \sigma|L)d\sigma$}$
> > > > >
> > > > > $=\pi(x) \sum_{L} \alpha(L)  \underbrace{\sum_{\substack{(\sigma, L) \in \Sigma(\mathcal{X}, \mathcal{N}):\\ \sigma_{0}=x, \textcolor{red}{\sigma_{L}=y}}} \Bigg[\overbrace{\Big(\prod_{l=1}^L Q_0(\sigma_{l-1}, \sigma_{l})\Big)}^\text{$T_1(\sigma|x, L)$}\overbrace{\min \left\\{ 1, \frac{\pi(y) \prod_{l=1}^L Q_0(\sigma_{l}, \sigma_{l-1})}{\pi(x) \prod_{l=1}^L Q_0(\sigma_{l-1}, \sigma_{l})} \right\\}}^\text{$A(x, y, \sigma|L)$}\Bigg]}_\text{$K(x,y|L) = \int_\sigma T_1(\sigma|x, L) T_2(y|x, \sigma, L) A(x, y, \sigma|L)d\sigma$}$
> > > > >
> > > > > When $\sigma_L \neq y$, the accept rate in the first equation is not well-defined. To avoid making the extra definition, we absorb the indicator $T_2$ as a constraint of the domain in integration in the second equation, hence $T_2$ does not explicitly show in our original equation.
> > > > >
> > > > > We are sorry for not making this clear enough in our draft, and we hope this explanation can answer your question. If you think it is now consistent, we will update the proof in our paper accordingly, following our discussion above.

---

> > > > ### Author Response · Authors · 2021-11-17
> > > > **Clarifying the transition kernel**
> > > >
> > > > Dear reviewer, thanks for your patience in discussion with us. In our sampler, as you mentioned, the auxiliary variable $u$ is the sampled auxiliary path $\sigma$. If we consider $\alpha(L)$ is dirac-delta, we have
> > > > - $T_1(u|x) = T_1(\sigma | x) = \prod_{l=1}^L Q_0(\sigma_{l-1}, \sigma_l)$
> > > > is the probability to sample the auxiliary path $\sigma$ condition on $x$.
> > > > - $T_2(y|x, u) = \mathbb{I}\\{x, y \text{ are the two ends of path } u (\text{or say} \sigma) \\}$
> > > > is deterministic, or say dirac-delta.
> > > >
> > > > Going back to our theorem, the equation, with a little abuse of notation, can be explained as:
> > > >
> > > > $\pi(x) \sum_{L} \alpha(L)  \underbrace{\sum_{\substack{(\sigma, L) \in \Sigma(\mathcal{X}, \mathcal{N}):\\ \sigma_{0}=x}} \underbrace{\Bigg[\overbrace{\Big(\prod_{l=1}^L Q_0(\sigma_{l-1}, \sigma_{l})\Big)}^\text{$T_1(\sigma|x, L)$} \overbrace{ \mathbb{I}\\{\sigma_L = y\\}}^\text{$T_2(y|x, \sigma, L)$} \overbrace{\min \left\\{ 1, \frac{\pi(y) \prod_{l=1}^L Q_0(\sigma_{l}, \sigma_{l-1}) \mathbb{I}\\{\sigma_0 = x\\} }{\pi(x) \prod_{l=1}^L Q_0(\sigma_{l-1}, \sigma_{l}) \mathbb{I}\\{\sigma_L = y\\} } \right\\}}^\text{$A(x, y, \sigma|L)$}\Bigg]}_\text{$T_1(\sigma|x, L) T_2(y|x, \sigma, L) A(x, y, \sigma|L)d\sigma$}}_\text{$K(x,y|L) = \int_\sigma T_1(\sigma|x, L) T_2(y|x, \sigma, L) A(x, y, \sigma|L)d\sigma$}$
> > > >
> > > > $=\pi(x) \sum_{L} \alpha(L)  \underbrace{\sum_{\substack{(\sigma, L) \in \Sigma(\mathcal{X}, \mathcal{N}):\\ \sigma_{0}=x, \textcolor{red}{\sigma_{L}=y}}} \Bigg[\overbrace{\Big(\prod_{l=1}^L Q_0(\sigma_{l-1}, \sigma_{l})\Big)}^\text{$T_1(\sigma|x, L)$}\overbrace{\min \left\\{ 1, \frac{\pi(y) \prod_{l=1}^L Q_0(\sigma_{l}, \sigma_{l-1})}{\pi(x) \prod_{l=1}^L Q_0(\sigma_{l-1}, \sigma_{l})} \right\\}}^\text{$A(x, y, \sigma|L)$}\Bigg]}_\text{$K(x,y|L) = \int_\sigma T_1(\sigma|x, L) T_2(y|x, \sigma, L) A(x, y, \sigma|L)d\sigma$}$
> > > >
> > > > When $\sigma_L \neq y$, the accept rate in the first equation is not well-defined. To avoid making the extra definition, we absorb the indicator $T_2$ as a constraint of the domain in integration in the second equation, hence $T_2$ does not explicitly show in our original equation.
> > > >
> > > > We are sorry for not making this clear enough in our draft, and we hope this explanation can answer your question. If you think it is now consistent, we will update the proof in our paper accordingly, following our discussion above.

---

> > > > > ### Comment · Reviewer_y5iJ · 2021-11-18
> > > > > **Thank you!**
> > > > >
> > > > > Thank you so much for the clarification! I am convinced that Theorem 1 is correct.
> > > > > I urge the authors to mention Auxiliary variable MCMC (cite equations from Liang et al) in the final paper to enable a novice to understand the claim. By default it is natural to assume that it is a marginal sampler.
> > > > >
> > > > > I have revealed in private discussions with other reviewers that my only issue was the fact that the proof of Theorem 1 was unclear to me and now that this is clarified, I believe the paper has significant merit.
> > > > > As such I update my score to Accept.
> > > > >
> > > > > I thank the authors for their patience and apologize if my tone in the original review was rude.
> > > > > I was solely interested in understanding the proof better and clearly there were steps in the proof that were not apparent to me at the outset.

---

> > > > > > ### Author Response · Authors · 2021-11-18
> > > > > > **Thanks!**
> > > > > >
> > > > > > Dear reviewer, thank you for your recognition of our work. We really appreciate your time in diving deeper into the discussion on the mathematical correctness of the sampler, which helps improve the rigor of the paper. We have made it clear that the proposed algorithm is a form of auxiliary sampler, and added our discussion about the auxiliary sampler in Appendix A.1 in the latest revision. Thanks again for your help in improving our work!

---

### Decision · Program_Chairs · 2022-01-20

**Decision:**

Accept (Spotlight)

**Comment:**

This paper provides a novel path auxiliary algorithm for more efficiently exploring discrete state spaces within a Metropolis-Hastings sampler for energy based models. In particular, it essentially replaces the "single site update" by instead proposing an entire path using local information, thus enabling the chain to take larger steps, which can improve acceptance/mixing significantly as they demonstrate. The work is a timely contribution that improves upon exciting recent work. After much discussion among several knowledgeable reviewers and clarifications regarding some details of the main theorem from the authors, there is consensus that the contributions are correct, novel, and likely of impact to the machine learning community. Since the revision period, the empirical evaluations have also been improved and the contributions have methodological novelty as well as promising practical performance.